# LEARNING A ZEROTH-ORDER OPTIMIZER FOR FINE-TUNING LLMS

## ABSTRACT

Zeroth-order optimizers have recently emerged as a practical approach for fine-tuning large language models (LLMs), significantly reducing GPU memory consumption compared to traditional first-order methods. Yet, existing zeroth-order methods rely on hand-crafted, static sampling strategies that are not adaptable to model-specific structures. To address this, we propose ZO Fine-tuner, a learning-based zeroth-order optimizer for LLMs that automatically learns efficient perturbation strategies through a compact and memory-efficient design. Crucially, our approach is motivated by the observation that only a small number of foundation models and their derivatives are widely adopted in practice. Therefore, learning the optimizer once for a given LLM and reusing it across diverse downstream tasks is both feasible and highly desirable. Accordingly, ZO Fine-tuner is designed to scale learning to learn (L2L) to the foundation-model era by supporting one-time training per LLM with minimal overhead. Experiments on 4 LLMs and 7 datasets show that ZO Fine-tuner outperforms prior zeroth-order baselines in 82.1% of task-model combinations, thereby demonstrating strong performance and scalability for efficient LLM fine-tuning.

## 1 INTRODUCTION

Nowadays fine-tuning pre-trained foundation models on downstream tasks has become a standard paradigm. However, as model sizes grow, traditional first-order optimizers such as Adam become increasingly expensive. In particular, these methods impose significant memory overhead, up to 12 times (Malladi et al., 2023) more than inference. Even with parameter-efficient fine-tuning (PEFT) methods such as LoRA (Hu et al., 2022) and Prefix-Tuning (Li & Liang, 2021), the overall memory requirement during training remains substantial.

To address these challenges, memory-efficient zeroth-order (MeZO) optimizer (Malladi et al., 2023) has been proposed. This approach only requires two forward passes per step and achieves competitive performance to first-order methods while maintaining memory usage comparable to inference. Many subsequent methods, such as HIZOO (Zhao et al., 2025), LOZO (Chen et al., 2024), MeZO-SVRG (Gautam et al., 2024), ZO-AdamU (Jiang et al., 2023), and ZO-DAP (Ma & Huang, 2025) attempt to improve upon MeZO by manually designing more sophisticated parameter-updating rules. However, these designs are often based on intuition or mathematical approximations, and still typically require extensive hyperparameter tuning beyond learning rates to perform well in practice.

We argue that prior works have largely overlooked the potential of learning to learn (L2L) techniques (Andrychowicz et al., 2016) in this context. Unlike hand-designed optimizers, L2L provides a data-driven approach to automatically learn effective optimization strategies. Rather than manually tuning update rules and hyperparameters, L2L leverages auxiliary neural networks that adaptively guide the optimization process. These learned optimizers typically rely on the same information accessible to conventional optimizers, such as gradient signals or their approximations. By leveraging such inputs, they often outperform their manually designed counterparts in both convergence speed and final performance, as they are able to explore the loss landscape more effectively during optimization (Wichrowska et al., 2017a). For example, learned optimizers have been shown to surpass SGD and even Adam across a variety of models and tasks (Lv et al., 2017a). Similar improvements have also been observed in zeroth-order optimization settings on small-scale models (Ruan et al., 2020).

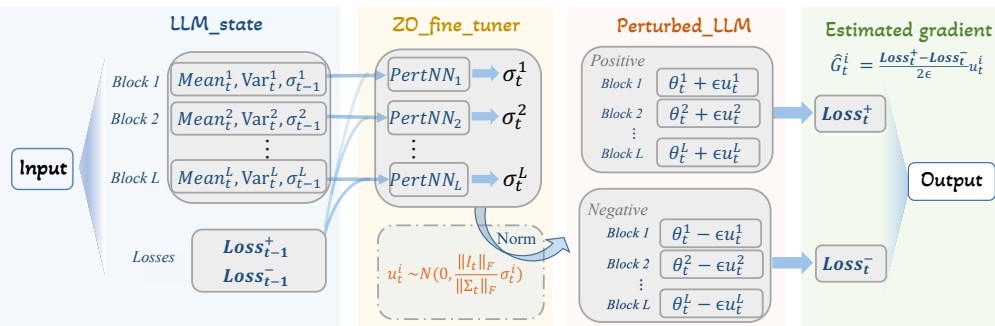

Figure 1: Fine-tune the LLM using trained ZO Fine-tuner. Each block of the LLM is equipped with a lightweight neural network that predicts its perturbation variance. For LLM parameter $\theta_t^i$ in block $i$ at step $t$, $\text{PertNN}_i$ takes in compact summarizing statistics containing the $\text{Mean}_t^i$, $\text{Var}_t^i$ of the $\theta_t^i$. Additionally, it takes in the last perturbation variance $\sigma_{t-1}^i$, and the two losses recorded at the last step. It outputs the updated perturbation variance $\sigma_t^i$ and then applies normalization. By learning non-uniform, layer-specific perturbation scales and plugging them into standard zeroth-order updates, the fine-tuner enables efficient, high-performance gradient-free optimization of LLM.

While L2L methods have shown promise on small-scale models (Chen et al., 2021), we believe their potential is even greater in the era of foundation models. In the small-model regime, different tasks typically require different models, and L2L optimizers often exhibit limited transferability across model architectures (Wichrowska et al., 2017a). As a result, a separate optimizer must be trained for each model-task pair, leading to substantial additional costs. In contrast, a recent LLM supply chain study shows that while there are many specialized checkpoints on platforms like Huggingface, most are derivatives of a handful of core base models like Llama and Qwen (Shahedur Rahman et al., 2025). Moreover, for a given LLM, the structure or properties leveraged by certain optimizers are often consistent across tasks (Guo et al., 2024). This provides a great opportunity for L2L methods, where **a learned optimizer trained once for a base LLM can be potentially reused across diverse derivative models and tasks**. Toward practical adoption, if model creators were to ship a pretrained learned finetuner alongside each base model, it would unlock a memory-efficient fine-tuning path with competitive performance for downstream users.

In the context of zeroth-order optimization for LLMs, learning a perturbation distribution with non-uniform and adaptive variance scales, rather than relying on a standard normal distribution, could be beneficial (Ye et al., 2018; Gao & Sener, 2022; Zhao et al., 2025). However, the sheer number of parameters of LLMs introduces new challenges when applying L2L as it requires differentiating through the optimization process itself, which demands storing a substantial number of activations for backpropagation. Moreover, naively applying coordinate-wise auxiliary networks at the LLM scale can result in prohibitive memory overhead. To address this, we draw inspiration from a careful theoretical analysis, which suggests that LLMs' approximately block-diagonal Hessian implies that sharing a single variance per block can already yield strong performance gains. We thus propose ZO Fine-tuner, which consists of highly compact and memory-efficient per-parameter-block auxiliary networks that learns shared effective perturbation variances. As a result, the **memory cost is minimal**: for OPT-30B, the total storage required for all auxiliary networks is less than 2MB under FP16 precision, which is negligible compared to the 60GB model itself. In the meantime, through extensive experiments, we demonstrate that our ZO Fine-tuner **trained on a single** dataset is **highly generalizable across model derivatives and datasets**, which strongly underscores the potential of the "train once, reuse widely" goal. Our contributions are summarized as follows:

• We extend L2L framework to LLMs and show that a single learned optimizer trained on a base model can generalize across downstream tasks and derivative checkpoints.

• Motivated by block-diagonal Hessian structure, we learn a shared perturbation variance per parameter block via compact per-block auxiliary networks. This dramatically reduced memory overhead compared to coordinate-wise or fully connected designs, which made L2L practical at LLM scale.

• Across four models and seven datasets (28 task-model pairs), ZO Fine-tuner outperforms the strongest baseline (lower training loss) in 82.1% of the combinations, achieving an average of 2.5% improvement in accuracy with tiny memory and time overhead.

## 2 RELATED WORK

**Zeroth-order optimization.** Zeroth-order optimization appears in a wide range of applications where either the objective function is implicit or its gradient is impossible or too expensive to compute. For example, methods (Tang et al., 2021; Hajinezhad & Zavlanos, 2018) consider derivative-free distributed algorithms for non-convex multi-agent optimization. ZO-BCD (Cai et al., 2021), ZOO (Chen et al., 2017), ZO-signSGD (Liu et al., 2019) and ZO-HessAware (Ye et al., 2019) utilize zeroth-order stochastic optimization to generate black-box adversarial example in deep learning. Beyond that, MeZO (Malladi et al., 2023) firstly adapted the classical ZO-SGD method to fine-tune LLMs, while achieving comparable performance with extremely great memory reduction. Subsequently, ZO-AdaMU (Jiang et al., 2023) improved ZO-SGD by incorporating momentum into its stochastic approximation process. HIZOO (Zhao et al., 2025) leverages Hessian information to enhance performance in a memory-efficient manner. Other works explore structural properties of the gradient to improve MeZO, such as utilizing low-rank approximations (Chen et al., 2024) or exploiting gradient sparsity (Guo et al., 2024; Liu et al., 2024).

**Learning to learn.** Previous studies have investigated using neural networks to improve optimization update rules, replacing manually crafted algorithms such as Adam (Kingma & Ba, 2015). (Cotter & Conwell, 1990) tried to use recurrent neural networks (RNNs) to model the optimization process to learn adaptively. After that, (Baxter, 1998) gave an overview of the idea and techniques of learning to learn; for example, they proposed to train RNNs to optimize basic convex functions. Then (Andrychowicz et al., 2016; Wichrowska et al., 2017b; Metz et al., 2019; 2022; Lv et al., 2017b) introduced a variety of sophisticated strategies to enhance the performance of optimizers in deep learning. Additionally, (Li & Malik, 2016) and (Li & Malik, 2017) adopted reinforcement learning (RL) policy search techniques into the L2L framework. In the context of zeroth-order optimization, (Ruan et al., 2020) applied L2L techniques to enhance performance on small-scale models.

## 3 METHOD

### 3.1 MOTIVATION

The foundational work in zeroth-order optimization for LLM fine-tuning, MeZO (Malladi et al., 2023), simply estimates the directional derivative as the step size on a certain sampled direction by evaluating the model at two perturbed parameter points. This approach only requires two forward passes and avoids backpropagation, making it attractive for memory-constrained training. Given a model with parameters $\theta \in \mathbb{R}^d$ and loss function $\mathcal{L}$, MeZO estimates the gradient on a mini-batch $\mathcal{B}_t$ as:

$$\hat{g}(\theta_t; \mathcal{B}_t) = \frac{\mathcal{L}(\theta_t + \epsilon u_t; \mathcal{B}_t) - \mathcal{L}(\theta_t - \epsilon u_t; \mathcal{B}_t)}{2\epsilon} u_t, \ \ u_t \sim \mathcal{N}(0, I_d), \tag{1}$$

and the model parameter is updated via $\theta_{t+1} = \theta_t - \eta \hat{g}(\theta_t; \mathcal{B}_t)$.

We argue that a fixed sampling rule from $\mathcal{N}(0, I)$ is suboptimal: the quality of zeroth-order gradient estimates depends on local properties of the landscape at each step (Ye et al., 2018; Gao & Sener, 2022; Zhao et al., 2025). Therefore, through learning, L2L approach has the potential to generate perturbations $u_t$ that are informed by such local signals and thus allocate perturbation effort more effectively. However, naive implementations of this idea can incur prohibitive memory overhead. For example, learning a separate perturbation for each individual parameter using a fully connected auxiliary network would require at least $O(d^2)$ parameters for a model with $d$ parameters.

We thus turn to exploit the geometric structure exhibited by LLM. Empirical evidence suggests that the Hessian of LLM is approximately block-diagonal, with blocks aligned to natural parameter groups (e.g., embeddings, attention Q,K,V matrices, projections, etc.) (Zhang et al., 2024b). This structure motivates a coarse control of the perturbation: rather than learning coordinate-wise perturbations, we target per-block adaptation. We now formalize this idea in theory, demonstrating that a simple adaptive variance change across parameter groups could lead to potential improvements over MeZO.

**Theorem 1** (Informal Version). *Define the expected change in loss after performing a one step update in parameter $\theta_t$ as $d(\theta_t) := \mathbb{E}\left[\mathcal{L}(\theta_{t+1}) \mid \theta_t\right] - \mathcal{L}(\theta_t)$. Suppose now the Hessian matrix $H(\theta_t)$ is block-diagonal $H(\theta_t) := \text{diag}(H_1(\theta_t), \cdots, H_b(\theta_t))$, then by varying the $\sigma_i$'s in $\Sigma := \text{diag}(\sigma_1 I, \cdots, \sigma_b I)$ the same gradient estimation equation 3.1 but with $u_t \sim \mathcal{N}(0, \Sigma)$ can yield tighter upper bound on $d(\theta_t)$ compared to MeZO.*

The formal version and proof of this theorem can be found in appendix E. At a high level, this theorem states that if the Hessian exhibits a block wise structure, then learning an adaptive per-block shared variance can improve convergence over MeZO. Crucially, the per-block parameterization yields this improvement *without* incurring prohibitive memory cost as the number of parameter blocks is far less than the number of parameters. For instance, in LLaMA-8B, the model contains only 291 parameter blocks, despite having over 8 billion individual parameters. This result thus motivates and justifies our design of ZO Fine-tuner, a per-block variance learner. Below, we first discuss the architecture of our ZO Fine-tuner and how to finetune downstream LLMs using a given ZO Fine-tuner. Then we introduce the training scheme for ZO Fine-tuner to enable generalizations.

## 3.2 ZO Fine-tuner

**Architecture.** As we discussed in the motivation section, we design ZO Fine-tuner to dynamically generate a block diagonal variance matrix $\Sigma_t$ corresponding to each parameter group at each optimization step via lightweight neural networks named PertNN. To incorporate all the dynamic information, PertNN takes in model parameters $\theta_t$, previously used perturbation variances $\Sigma_{t-1}$, and their observed loss as inputs $\ell_{t-1}$. Intuitively, these inputs encourage PertNN to consider the effectiveness of past updates, where perturbations that lead to sharper loss changes might indicate more informative directions. However, we notice that the model parameters are still too memory-intensive as an input feature. Therefore, we further compress the memory usage by only feeding the summarizing statistics of the model parameters $\theta_t$ into PertNN, such as $\text{Mean}(\theta_t)$ and $\text{Var}(\theta_t)$.

Formally, the perturbation variance at each step $t$ is generated as follows. For parameter block $i$, $\sigma_{t-1}^{(i)}$ is the previous perturbation variance, $d_i$ is the number of parameters in this block, and $\text{Mean}_t^{(i)}$ and $\text{Var}_t^{(i)}$ represent the current mean and variance of the block's parameter values. $\omega^{(i)}$ denotes the learnable parameters of the auxiliary neural network assigned to block $i$.

$$\sigma_t^{(i)} = \text{PertNN}^{(i)}\left(\ell_{t-1}, \sigma_{t-1}^{(i)}, \text{Mean}_t^{(i)}, \text{Var}_t^{(i)}; \omega^{(i)}\right),$$
$$\Sigma_t = \text{diag}(\sigma_t^{(1)} I_{d_1}, \sigma_t^{(2)} I_{d_2}, \ldots, \sigma_t^{(n)} I_{d_n}). \tag{2}$$

With this variance, ZO Fine-tuner then updates model parameters with

$$\hat{g}(\theta_t; \mathcal{B}_t; \omega) := \frac{\mathcal{L}(\theta_t + \epsilon u_t; \mathcal{B}_t) - \mathcal{L}(\theta_t - \epsilon u_t; \mathcal{B}_t)}{2\epsilon} u_t, \quad u_t \sim \mathcal{N}(0, \Sigma_t)$$
$$\theta_{t+1} := \theta_t - \eta\, \hat{g}(\theta_t; \mathcal{B}_t) \tag{3}$$

Importantly, we should note that $\hat{g}$ is inherently a function of $u_t$, which is a function of $\Sigma_t$, and thus a function of the parameter of PertNN $\omega$. To enable gradient-based training of PertNN within the L2L framework, we adopt the reparameterization trick: instead of sampling $u_t$ directly from $\mathcal{N}(0, \Sigma_t)$, we sample $z_t \sim \mathcal{N}(0, I_d)$ and compute $u_t = \Sigma_t^{1/2} z_t$. This makes the entire perturbation process differentiable, allowing gradients to flow back through the perturbation generation module.

**Normalization.** Although effective, this non-uniform variance introduced a new challenge when using ZO Fine-tuner as an optimizer. From the two-point ZO estimator, we see that

$$\mathbb{E}[\hat{g}(\theta_t; \mathcal{B}_t)] = \mathbb{E}[\frac{\mathcal{L}(\theta_t + \varepsilon u_t; \mathcal{B}_t) - \mathcal{L}(\theta_t - \varepsilon u_t; \mathcal{B}_t)}{2\varepsilon}] \approx \mathbb{E}[u_t u_t^\top]\nabla\mathcal{L}(\theta_t; \mathcal{B}_t). \tag{4}$$

Therefore, when fine-tuning downstream tasks, we note that the effective learning rate became $\eta \cdot \frac{\|u_t\|^2}{d}$ on average. This makes controlling the effective learning rate difficult, and the learned variance $\Sigma_t$ became a confounding variable in the update size. In reality, we wish $\Sigma_t$ to only carry information about relative block-wise variance, and we could still use a single learning rate to control the overall step size to ensure stable training. Therefore, we introduce the following normalization, which ensures the decoupling of the variance and the learning rate. We note that if $u_t = \Sigma_t^{1/2} z_t$ with $z_t \sim \mathcal{N}(0, I)$, then $\mathbb{E}\|u_t\|^2 = \text{tr}(\Sigma_t \Sigma_t^\top) = \|\Sigma_t\|_F^2$. We then normalize by fixing the total variance budget and let $\|\Sigma_t\|_F^2 = \|I_d\|_F^2 = d$. Thus, only the relative block-wise variances are learned. In practice, this keeps $\|u_t\|$ approximately constant (by concentration in high dimensions). For example, with our generated $\Sigma_t$, if $u_t \sim \mathcal{N}(0, \Sigma_t)$, $\|u_t\|$ concentrates around $\|\Sigma_t\|_F$ and we achieve the desired control over the effective learning rate.

**Complete Optimization Algorithm.** As summarized in Algorithm 1 and Figure 1, ZO Fine-tuner first compute the block-wise non-uniform perturbation variance $\Sigma_t$ using the learned neural network PertNN. Then it applies normalization to control the overall magnitude of the perturbation. Finally, it uses the normalized perturbation to update the LLM following equation 3. We notice this incurs minimal overhead compared to MeZO, in terms of both memory and speed. In particular, the only memory overhead compared to MeZO is the light-weight per-block PertNN, whereas the only speed overhead is the query to PertNN.

---

**Algorithm 1** Finetuning a LLM with ZO Fine-tuner

---

**Require:** LLM parameters $\theta$, PertNN parameters $\omega$, training step $T$, learning rate $\eta$
1: Initialize variance $\Sigma_0$ as $I_d$, LLM parameter as $\theta_0$.
2: **for** $t = 1, ..., T$ **do**
3:     Sample a batch $\mathcal{B}_t$ from $\mathcal{T}$
4:     $\Sigma_t \leftarrow \text{PertNN}(\theta_t, \Sigma_{t-1}, \boldsymbol{\ell}_{t-1}; \omega)$
5:     Sample $u_t \sim N(0, \frac{\|I_d\|_F}{\|\Sigma_t\|_F}\Sigma_t)$                    ▷ Sample after normalization
6:     Compute LLM loss with perturbed parameter to obtain $\ell_t$
7:     $\hat{g}_t = \frac{l_t^+ - l_t^-}{2\epsilon}u_t$
8:     $\theta_{t+1}^t = \theta_t - \eta\hat{g}_t$
9: **end for**

---

### 3.3 TRAINING ZO FINE-TUNER

We now turn to training ZO Fine-tuner in a L2L fashion. The key idea is to treat the model's own finetuning trajectory as supervision. After a single update by ZO Fine-tuner, we evaluate the post-update loss and adjust PertNN so as to reduce this quantity across tasks. We next formalize this meta-objective and outline several practical choices that make training stable.

**Data Source and Objective Function.** First, we need a source of training data for our ZO Fine-tuner. In our setting, this data corresponds to different model states with various losses. A key insight of us is to notice that the fine-tuning process of LLMs under a first-order optimizer naturally produces a trajectory of intermediate model states, and we can directly leverage this trajectory to optimize the perturbation variance generator.

Along the first order optimization trajectory with loss function $\mathcal{L}$, we obtain a set of model parameters $\{\theta_0^k\}_k$. We then attempt to perform a one-step zeroth-order update using our ZO Fine-tuner with update rule 3 to get $\theta_1^k$ and use the resulting loss as a feedback signal to assess and optimize the effectiveness of the current perturbation strategy. Specifically, at each step we aim to minimize the post-update loss $\mathcal{L}(\theta_1^k)$. As we discussed in section 3.2, the estimated gradient $\hat{g}$ is implicitly a differentiable function of the parameters $\omega$ of PertNN per the reparametrization trick. Therefore, we can use a gradient-based method to update ZO Fine-tuner. Formally, the objective for training ZO Fine-tuner is therefore:

$$\min_\omega \mathcal{L}_{\text{ZO}}(\theta_0^k; \omega) := \min_\omega \mathcal{L}(\theta_1^k) = \min_\omega \mathcal{L}\left(\theta_0^k - \eta\,\hat{g}(\theta_0^k, \omega)\right) \tag{5}$$

After the update, we move the parameters $\theta_0^k$ along the first-order trajectory to get $\theta_0^{k+1}$ and continue learning. As the inputs to ZO Fine-tuner are task and model-agnostic state summaries, rather than task-specific features, the learned decisions are largely invariant to differences across datasets or nearby checkpoints. As we will demonstrate in experiments, our ZO Fine-tuner trained on one single dataset can be transferred to efficiently finetune other datasets and model derivatives.

**Periodic Reset of Model Parameters.** During the training of our ZO Fine-tuner, a lot of data needs to be generated. However, since the optimizer is trained along the fine-tuning trajectory of a model using a first-order optimizer, the auxiliary network tends to receive inputs that are chronologically ordered. In particular, it will get more data from the low-loss region. As a result, it may lead to overfitting to the low-loss region of the parameters while learning the crucial high-loss region insufficiently.

To address this issue, we introduce a periodic re-initialization mechanism. After each complete optimization cycle or when the loss has sufficiently decreased, we reset the model parameters to their original pre-finetuning state and restart the fine-tuning process. This approach introduces diversity into the input distribution by exposing the optimizer to model states from multiple phases of training.

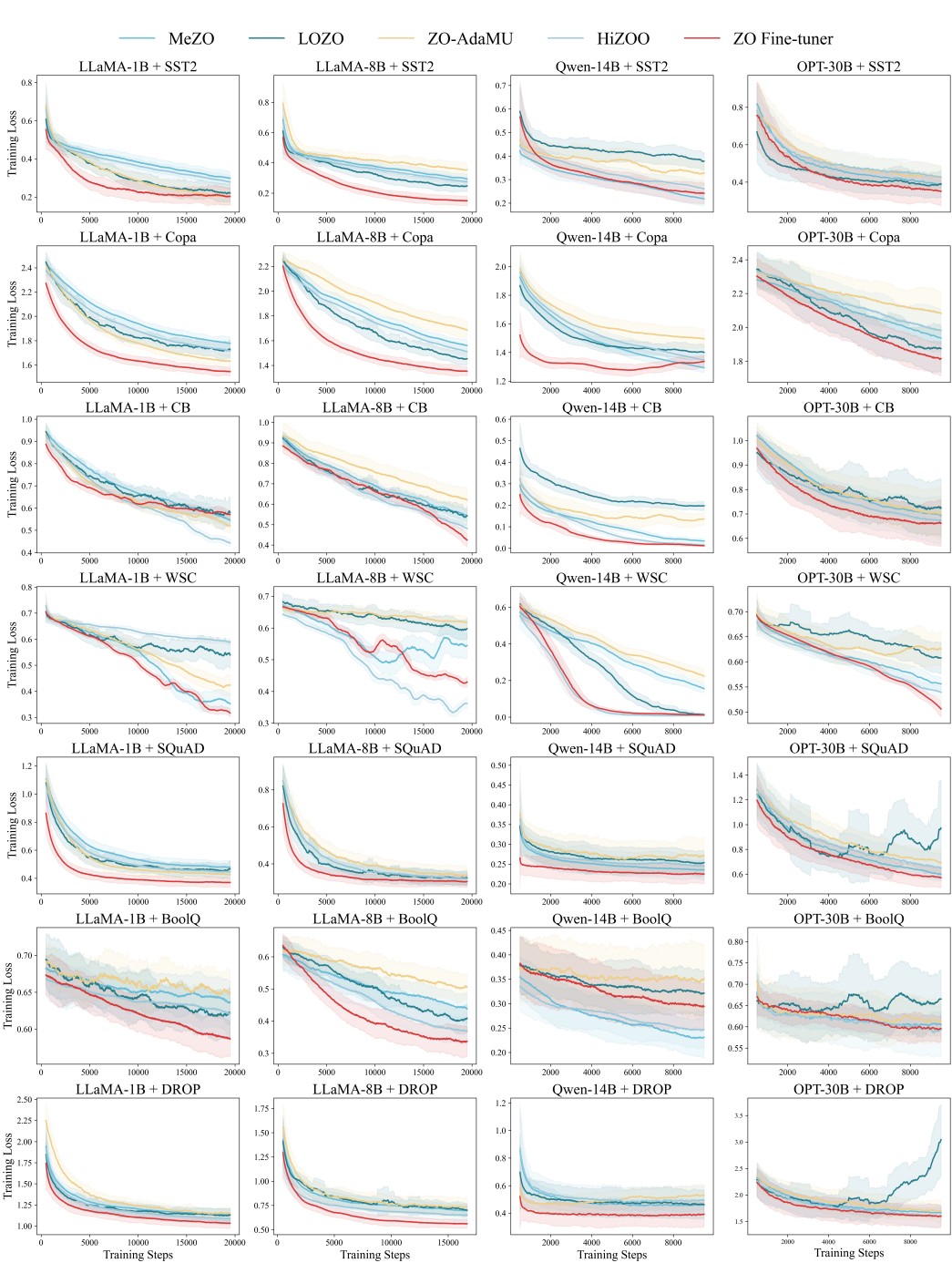

Figure 2: Loss comparison across different methods on various datasets and LLMs. Models (columns) are LLaMA-3.2-1B, LLaMA-3.1-8B, Qwen2.5-14B and OPT-30B, while datasets (rows) cover COPA, SST-2, CB, SQuAD, WSC, BoolQ and DROP. All curves use the best hyperparameters found for each method. The shaded region around each curve shows the standard deviation of the smoothed loss—the wider the shade, the larger the fluctuation. ZO Fine-tuner shows advantages in both convergence speed and final loss value across most settings.

---

**Algorithm 2** Learning to Learn Framework

---

**Require:** LLM parameters $\theta$, training step $T$, loss function for LLM $\mathcal{L}$, learning rate for LLM $\eta_1$,
   learning rate for PertNN $\eta_2$, task list $\mathcal{T}_{\text{list}}$, perturbation scale $\epsilon$.

1: Initialize PertNN as $\omega_0$, LLM parameter as $\theta_0$, perturbation variance $\Sigma_0^{\mathcal{T}}$ as $I_d$ for each task $\mathcal{T}$.
2: **for** $t = 1, ..., T$ **do**
3:     $\mathcal{T}_{\text{list}} \leftarrow \text{Shuffle}(\mathcal{T}_{\text{list}})$
4:     **for** each task $\mathcal{T}$ in $\mathcal{T}_{\text{list}}$ **do**
5:         Sample a batch $\mathcal{B}_t^{\mathcal{T}}$ from $\mathcal{T}$
6:         $\Sigma_t^{\mathcal{T}} \leftarrow \text{PertNN}(\theta_t, \Sigma_{t-1}^{\mathcal{T}}, \ell_{t-1}^{\mathcal{T}}; \omega_t)$; normalize such that $\|\Sigma_t^{\mathcal{T}}\|_F^2 = \|I_d\|_F^2$
7:         Sample $u_t \sim \mathcal{N}(0, \Sigma_t)$ and Compute LLM loss with perturbed parameter to obtain $\ell_t$
8:         $\mathcal{L}_{\text{ZO}} \leftarrow \mathcal{L}^{\mathcal{T}}(\theta_t - \eta_1 \frac{l_t^+ - l_t^-}{2\epsilon} u_t; \mathcal{B}_t^{\mathcal{T}})$, $\omega_t \leftarrow \omega_t - \eta_2 \frac{\partial \mathcal{L}_{\text{ZO}}}{\partial \omega_t}$     ▷ Update PertNN with SGD
9:         $l_t \leftarrow \mathcal{L}^{\mathcal{T}}(\theta_t; \mathcal{B}_t^{\mathcal{T}})$, $\theta_t \leftarrow \theta_t - \eta_1 \frac{\partial l_t}{\partial \theta_t}$                          ▷ Update LLM with SGD
10:     **end for**
11:     $\omega_{t+1} \leftarrow \omega_t, \theta_{t+1} \leftarrow \theta_t$
12:     When the training step $t$ reaches a predefined period, reinitialize LLM parameter as $\theta_0$.
13: **end for**

---

**Complete Learning to Learn Framework for ZO Fine-tuner Training.** Algorithm 2 illustrates the complete L2L framework for training ZO Fine-tuner. At each training step, we sample a training dataset and a batch from this dataset to perform a one-step update to ZO Fine-tuner as described above. Moreover, we periodically reset the model parameters to mitigate the bias discussed previously. Despite the complexity of this training algorithm, we would like to emphasize that it is a *one-time* cost: once ZO Fine-tuner is learned, deployment reduces to Algorithm 1.

## 4  EXPERIMENT

Following MeZO Malladi et al. (2023), we evaluate ZO Fine-tuner with four LLMs: LLaMA-3.2-1B (Grattafiori et al., 2024), LLaMA-3.1-8B (Grattafiori et al., 2024), Qwen2.5-14B (Bai et al., 2023), and OPT-30B (Zhang et al., 2022) using seven diverse benchmark datasets including SST-2 (Socher et al., 2013), CB (De Marneffe et al., 2019), COPA (Roemmele et al., 2011), BoolQ (Clark et al., 2019), WSC (Levesque et al., 2012), SQuAD (Rajpurkar et al., 2016), and DROP (Dua et al., 2019).

We compare our approach against four representative zeroth-order optimization baselines for LLM fine-tuning: HIZOO (Zhao et al., 2025), LOZO (Chen et al., 2024), MeZO and MeZO-Adam (Malladi et al., 2023). Due to computational resource constraints, we replace the expensive MeZO-Adam with a more efficient variant MeZO-AdamU (Jiang et al., 2023) for models larger than LLaMA-3.2-1B. To ensure a fair comparison, we perform the same grid search over learning rates for each method and pick the best learning rate when reporting. More details can be found in appendix C.2.

For our ZO Fine-tuner, we train it once using algorithm 2 on the COPA dataset. This choice is mainly due to COPA's consistently smooth loss decrease during standard fine-tuning. Its small size and also yield fast cycles. **Unless otherwise noted, the ZO Fine-tuner trained on COPA is reused as is across all other tasks and models.** In appendix D.5, we also discussed more about multi-dataset training. Other hyperparameters and training details can be found in section C.3.

### 4.1  MAIN RESULTS

**Generalization Across Datasets.** Figure 2 compares convergence across all 28 dataset-model pairs using each method's best learning rate. We observe that ZO Fine-tuner (red) consistently reaches lower loss faster. The effect is especially clear on SST-2, CB, COPA, SQuAD, and DROP, where curves descend more steeply early on and settle at a better plateau. In addition, we report the final loss and accuracy values for all 28 combinations in Table 1. On average, ZO Fine-tuner achieves an average accuracy improvement of 2.5% over MeZO. Overall, Our method outperforms the baselines in 75.0% of the task-model combinations in accuracy and 82.1% in the converged loss. These results indicate strong generalization capability of ZO Fine-tuner, as training ZO Fine-tuner on a single COPA dataset already yields consistent gains across datasets.

Table 1: Average training loss in the final epoch and accuracy on seven datasets for each method and model combination under the best hyperparameter. We report both loss (↓) and accuracy / F1 (↑) across tasks of diverse formats to evaluate the overall performance.

| Model | Method | COPA | | SST-2 | | CB | | SQuAD | | WSC | | BoolQ | | DROP | |
|---|---|---|---|---|---|---|---|---|---|---|---|---|---|---|---|
| | | Loss | Acc | Loss | Acc | Loss | Acc | Loss | F1 | Loss | Acc | Loss | Acc | Loss | F1 |
| LLaMA-3.2-1B | MeZO | 1.77 | 0.75 | 0.29 | 0.90 | 0.55 | 0.70 | 0.48 | 0.75 | 0.35 | **0.62** | 0.63 | 0.63 | 1.16 | 0.29 |
| | MeZO-Adam | 1.62 | 0.79 | 0.20 | 0.92 | 0.53 | 0.66 | 0.41 | **0.78** | 0.42 | 0.61 | 0.66 | 0.62 | 1.14 | 0.29 |
| | HIZOO | 1.71 | 0.78 | 0.27 | 0.90 | **0.44** | **0.71** | 0.43 | 0.75 | 0.55 | 0.54 | 0.62 | 0.61 | 1.09 | 0.29 |
| | LOZO | 1.72 | 0.74 | 0.20 | 0.92 | 0.58 | 0.64 | 0.47 | **0.78** | 0.51 | 0.61 | 0.62 | 0.64 | 1.15 | 0.32 |
| | ZO Fine-tuner | **1.54** | **0.80** | **0.14** | **0.93** | 0.57 | 0.67 | **0.37** | **0.78** | **0.31** | 0.56 | **0.58** | **0.66** | **1.03** | **0.35** |
| LLaMA-3.1-8B | MeZO | 1.54 | 0.92 | 0.29 | 0.92 | 0.54 | 0.71 | 0.32 | 0.89 | 0.55 | 0.63 | 0.42 | 0.78 | 0.69 | 0.64 |
| | MeZO-AdamU | 1.67 | 0.89 | 0.36 | 0.92 | 0.61 | 0.70 | 0.35 | 0.86 | 0.61 | **0.64** | 0.50 | 0.75 | 0.73 | 0.59 |
| | HIZOO | 1.50 | **0.93** | 0.27 | 0.92 | 0.47 | 0.71 | 0.32 | 0.88 | **0.36** | 0.62 | 0.36 | 0.79 | 0.64 | 0.60 |
| | LOZO | 1.46 | 0.89 | 0.25 | **0.94** | 0.54 | 0.70 | 0.33 | **0.90** | 0.61 | 0.63 | 0.41 | 0.83 | 0.74 | 0.65 |
| | ZO Fine-tuner | **1.35** | 0.91 | **0.18** | **0.94** | **0.26** | **0.76** | **0.31** | **0.90** | 0.44 | 0.62 | **0.34** | **0.87** | **0.54** | **0.66** |
| Qwen2.5-14B | MeZO | 1.28 | 0.86 | **0.21** | 0.88 | 0.05 | **0.93** | 0.24 | 0.88 | 0.18 | 0.76 | **0.23** | 0.84 | 0.45 | 0.66 |
| | MeZO-AdamU | 1.43 | 0.85 | 0.35 | 0.89 | 0.13 | 0.91 | 0.28 | 0.90 | 0.25 | 0.75 | 0.35 | 0.84 | 0.50 | 0.64 |
| | HIZOO | **1.34** | 0.87 | 0.26 | 0.93 | **0.03** | 0.89 | 0.24 | 0.89 | **0.02** | **0.79** | 0.25 | 0.86 | 0.49 | 0.68 |
| | LOZO | 1.40 | 0.91 | 0.38 | 0.93 | 0.19 | 0.91 | 0.26 | 0.90 | 0.04 | **0.79** | 0.32 | 0.86 | 0.46 | 0.67 |
| | ZO Fine-tuner | **1.34** | **0.92** | 0.24 | **0.94** | **0.03** | **0.93** | **0.22** | **0.91** | **0.02** | 0.76 | 0.29 | **0.89** | **0.40** | **0.70** |
| OPT-30B | MeZO | 1.93 | 0.83 | 0.38 | 0.89 | 0.69 | 0.64 | 0.59 | 0.74 | 0.55 | **0.63** | **0.60** | 0.66 | 1.66 | **0.31** |
| | MeZO-AdamU | 2.07 | 0.80 | 0.43 | 0.84 | 0.70 | 0.66 | 0.67 | 0.73 | 0.62 | **0.63** | 0.62 | 0.66 | 1.70 | 0.30 |
| | HIZOO | 1.97 | 0.81 | 0.43 | 0.86 | 0.67 | 0.66 | 0.65 | 0.75 | 0.53 | 0.61 | 0.62 | 0.65 | 1.61 | 0.30 |
| | LOZO | 1.86 | 0.82 | 0.40 | **0.90** | 0.73 | 0.64 | 0.96 | 0.75 | 0.58 | 0.62 | 0.70 | 0.66 | 2.83 | 0.27 |
| | ZO Fine-tuner | **1.81** | **0.85** | **0.35** | 0.87 | **0.66** | **0.70** | **0.56** | **0.77** | **0.51** | 0.60 | 0.61 | **0.67** | **1.59** | 0.31 |

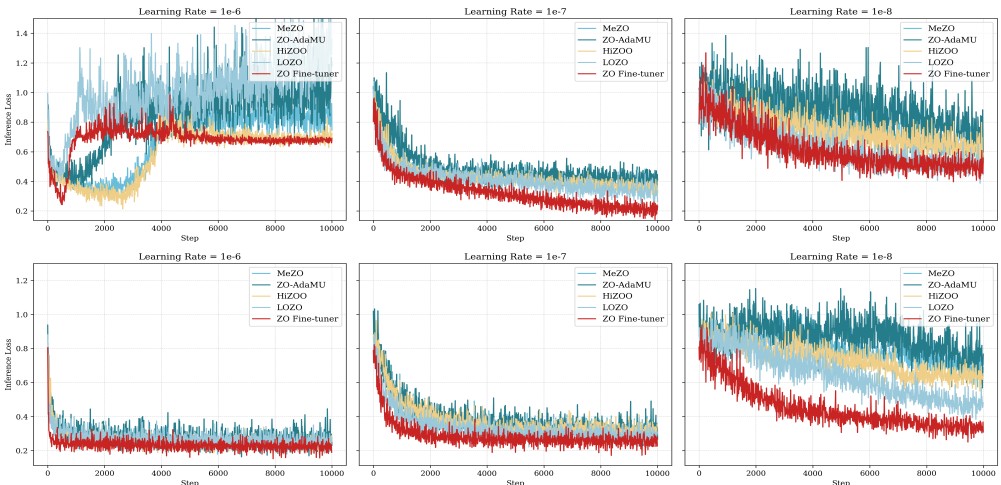

Figure 3: Loss curves under varying learning rates for different optimizers on (top) SST2 with LLaMA-3.1-8B, and (bottom) SQuAD with Qwen2.5-14B.

**Generalization Across Model Derivatives.** We further investigate the generalization capability of ZO Fine-tuner to derived models. We take the ZO Fine-tuner trained with LLaMA-3.1-8B and use it to finetune Llama-3.1-8B-Instruct. As can be seen in table 3, ZO Fine-tuner can also generalize to effectively finetune derived models across a single model family. On both datasets evaluated, ZO Fine-tuner beats MeZO both in terms of both average loss and final accuracy. Practically, this supports the *train-once, reuse-across-derivatives* paradigm we have mentioned. If model developers could release a pretrained finetuner with each base model, then model users can then efficiently finetune the model further on derivative checkpoints with near-inference memory.

## 4.2 ABLATION STUDIES

**Learning Rate.** Figure 3 further demonstrates the sensitivity of different methods to the choice of learning rate. Notably, ZO Fine-tuner often achieves comparable loss at a learning rate of $1 \times 10^{-8}$ to

Table 2: Ablation results on Normalization and Periodic Reset. We report the final loss and the final accuracy. Consistently, both techniques individually improve performance across models and datasets, and combining them achieves the best results.

| Setting | LLaMA-8B + SQuAD | LLaMA-8B + SST2 | Qwen-14B + SQuAD | Qwen-14B + SST2 |
|---|---|---|---|---|
| Base | 0.3950 / 0.840 | 0.3976 / 0.874 | 0.3582 / 0.844 | 0.4086 / 0.800 |
| Reset alone | 0.3682 / 0.856 | 0.3891 / 0.881 | 0.3551 / 0.851 | 0.4039 / 0.810 |
| Normalization alone | 0.3071 / 0.899 | 0.3061 / 0.920 | 0.2380 / 0.904 | 0.3885 / 0.844 |
| Reset+Normalization | **0.3065 / 0.905** | **0.1789 / 0.941** | **0.2246 / 0.911** | **0.2403 / 0.935** |

Table 3: We demonstrate that the ZO Fine-tuner trained form LLaMA-3.1-8B generalizes well to LLaMA-3.1-8B-Instruct. Across datasets it outperforms MeZO in final loss and accuracy.

| Method | Dataset | Loss / Acc |
|---|---|---|
| SST2 | MeZO | 0.276 / 0.92 |
| | ZO Fine-tuner | **0.164 / 0.95** |
| SQuAD | MeZO | 0.291 / 0.90 |
| | ZO Fine-tuner | **0.287 / 0.92** |

Table 4: Ablation results on parameter sharing strategy. We compare our block-wise scheme to a simpler layer-wise baseline. As shown below, block-wise sharing consistently achieves lower final loss and higher accuracy.

| Model | Sharing | SST2 Loss / Acc | SQuAD Loss / Acc |
|---|---|---|---|
| LLaMA-8B | layer wise | 0.23 / 0.92 | 0.32 / 0.88 |
| | block wise | **0.18 / 0.94** | **0.31 / 0.90** |
| Qwen-14B | layer wise | 0.27 / 0.91 | 0.25 / 0.88 |
| | block wise | **0.24 / 0.94** | **0.22 / 0.91** |

that of baseline methods operating at $1 \times 10^{-7}$. When the learning rate further increases to $1 \times 10^{-6}$, many baseline methods suffer from instability and fail to converge, falling short of the performance that ZO Fine-tuner achieves at $1 \times 10^{-7}$. More results can be found in D.1.

**Normalization & Periodic Reset.** We also conduct experiments to evaluate the effectiveness of our design choices including normalization introduced in section 3.2 and periodic reset in section 3.3. From table 2, it is clear that both normalization and periodic reset helps ZO Fine-tuner for achieving better performance.

**Parameter Sharing Strategy.** Finally, we evaluate the granularity of sharing in ZO Fine-tuner, comparing our *block-wise* scheme to a simpler *layer-wise* sharing baseline. As shown in Table 4, block-wise sharing consistently achieves lower final loss and higher accuracy. Importantly, this choice is theory-driven: when the Hessian is (approximately) block-diagonal, theorem 1 indicates that the natural unit for variance sharing is the Hessian block itself.

### 4.3 MEMORY USAGE AND TIME EFFICIENCY ANALYSIS

**Memory Usage.** The memory overhead of ZO Fine-tuner when using to finetune LLMs mainly comes from the additional memory taken by PertNN. However, the parameter number of our ZO Fine-tuner is extremely small, even negligible compared to the LLMs. Consequently, the memory footprint of our method remains essentially identical to that of MeZO. Under equivalent experimental settings, it requires only $1/4$ of the memory overhead incurred by Adam. For example, MeZO and ZO Fine-tuner peak at 61GB and 62GB of GPU memory when fine-tuning OPT-30B, whereas first-order Adam reaches 312GB with FP16. More details can be found in D.3.

**Time Efficiency.** Similarly, the time overhead comes directly from the query to PertNN. However, this overhead is typically minimal. For example, when fine-tuning on DROP using LLaMA-3.2-1B on an L40S GPU with a batch size of 16, the generation of perturbation takes only 0.025 seconds, while all other operations take approximately 0.70 seconds. This means our method introduces less than 3.4% additional overhead, demonstrating strong time efficiency. This overhead becomes even less significant with larger models. For instance, under the same setting with LLaMA-3.1-8B, perturbation generation takes only 0.052 seconds compared to a total runtime of 3.14 seconds. More details can be found in D.2.

## 5 CONCLUSION

We introduced ZO Fine-tuner, a learning-to-learn zeroth-order optimizer that uses adaptive, per-block perturbation variances. The finetuner trained once on a single dataset is demonstrated to transfer across tasks and to finetuned derivatives, supporting a practical "train once, reuse widely" path.

## ETHICS STATEMENT

This work does not raise any known ethical concerns, as far as the authors concern.

## REPRODUCIBILITY STATEMENT

Our code repository is available at `https://anonymous.4open.science/r/ZO_Fine_tuner_ICLR-F69A`. We have also more detailedly discuss the hyperparameters and experiment setting in appendix C.2 and C.3.

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

## A  LLM Usage

We used LLM only to polish writing and retrieve related works for this work.

## B  Discussions

### B.1  Limitations and Future Work

The coordinate-wise structure has already been shown to be effective and reasonable in the era of L2L for zeroth-order optimizer on small-scale models (Ruan et al., 2020). While our current design uses a diagonal variance matrix $\Sigma_t$ for its memory efficiency and strong empirical performance, exploring non-diagonal structures is a potential improvement, though it may require additional techniques to mitigate the associated memory overhead.

Moreover, more properties of LLM gradient and Hessian could be potentially exploited. For example, Chen et al. (2024); Sun et al. (2025) explicitly exploits the low-rank structure of LLM gradients. A potential future direction is to leverage these properties to generate more informed perturbations or cut the memory usage even more.

## C  Implementation Details

### C.1  Datasets and Models

We evaluate all optimizers on seven NLP tasks spanning multiple formats, including natural language inference, question answering, and commonsense reasoning. SST-2 (Socher et al., 2013) is a binary sentiment classification benchmark from the GLUE suite. CB (De Marneffe et al., 2019) and COPA (Roemmele et al., 2011) are low-resource natural language inference tasks from SuperGLUE, requiring models to recognize textual entailment or choose causal relationships. BoolQ (Clark et al., 2019) involves answering yes/no questions given short passages. WSC (Levesque et al., 2012) tests pronoun resolution in challenging coreference contexts. SQuAD (Rajpurkar et al., 2016) and DROP (Dua et al., 2019) are span-based question answering datasets that require locating answer spans in context paragraphs. For most classification tasks, we report accuracy as the evaluation metric. For SQuAD and DROP, we follow standard practice and report F1 score to better capture partial match quality.

We evaluate our optimizers on four representative large language models with diverse architectures and scales: LLaMA-3.2-1B (Grattafiori et al., 2024), LLaMA-3.1-8B (Grattafiori et al., 2024), Qwen2.5-14B (Bai et al., 2023), and OPT-30B (Zhang et al., 2022).

### C.2  Hyperparameters

We use a two-layer MLP with 64 hidden units and a tanh activation function as the auxiliary neural network for each parameter block. Table 5 presents the hyperparameter search grids used in our experiments to facilitate reproducibility. We primarily perform a grid search over three learning rate values: $10^{-4}$, $10^{-5}$, and $10^{-6}$ for MeZO-Adam, and $10^{-6}$, $10^{-7}$, and $10^{-8}$ for all other methods. For the WSC task, we additionally include $3 \times 10^{-7}$, as most methods exhibit slow loss decay at $10^{-7}$ and become unstable when using $10^{-6}$. We run 20,000 optimization steps on LLaMA-1B and LLaMA-8B, and 10,000 steps on Qwen-14B and OPT-30B due to resource limitation. A batch size of 16 is used for all models by default, except for OPT-30B, where we reduce it to 4 due to GPU memory constraints. Also due to computational resource constraints, we replace the expensive MeZO-Adam with its more efficient variant MeZO-AdamU (Jiang et al., 2023) for models larger than LLaMA-1B. As shown in the hyperparameter table, our method, together with MeZO, requires the smallest number of tunable hyperparameters among all baselines.

### C.3  Learning to Learn Details

In Section 3.3, we introduced our learning to learn framework. Here, we elaborate on additional implementation details. We use a two-layer MLP with 64 hidden units and a tanh activation function as the auxiliary neural network for each parameter block. Empirically, we set $\epsilon = 10^{-3}$, $\eta_1 = 10^{-6}$, and $\eta_2 = 10^{-2}$ in Algorithm 2. When the task list $\mathcal{T}_{\text{list}}$ contains only a single task, the framework

Table 5: Hyperparameter configurations for ZO Fine-tuner and all baseline methods.

| Method | Hyperparameters | Values |
|---|---|---|
| MeZO | Batch size | 16 for LLaMA-1B/8B/Qwen-14B; 4 for OPT-30B |
| | Learning rate | $\{10^{-6}, 10^{-7}, 10^{-8}\}$ (plus $3 \times 10^{-7}$ for WSC only) |
| | $\epsilon$ | $10^{-3}$ |
| MeZO-Adam | Batch size | 16 |
| | Learning rate | $\{10^{-4}, 10^{-5}, 10^{-6}\}$ (plus $3 \times 10^{-6}$ for WSC only) |
| | $\epsilon$ | $10^{-3}$ |
| | $\epsilon_{\text{Adam}}$ | $\{10^{-6}, 10^{-7}, 10^{-8}\}$ |
| ZO-AdamU | Batch Size | 16 for LLaMA-1B/8B/Qwen-14B; 4 for OPT-30B |
| | Learning Rate | $\{10^{-6}, 10^{-7}, 10^{-8}\}$ (plus $3 \times 10^{-7}$ for WSC only) |
| | $\epsilon$ | $10^{-3}$ |
| | $\alpha$ | $\{0.2, 0.5, 0.7\}$ |
| | $\beta^{(1)}$ | $\{0.9, 0.8, 0.7\}$ |
| | $\beta^{(2)}$ | $\{0.01, 0.05, 0.1\}$ |
| HIZOO | Batch Size | 16 for LLaMA-1B/8B/Qwen-14B; 4 for OPT-30B |
| | Learning Rate | $\{10^{-6}, 10^{-7}, 10^{-8}\}$ (plus $3 \times 10^{-7}$ for WSC only) |
| | $\epsilon$ | $10^{-3}$ |
| | Smooth Constant | $\{10^{-7}, 10^{-8}\}$ |
| LOZO | Batch Size | 16 for LLaMA-1B/8B/Qwen-14B; 4 for OPT-30B |
| | Learning Rate | $\{10^{-6}, 10^{-7}, 10^{-8}\}$ (plus $3 \times 10^{-7}$ for WSC only) |
| | $\epsilon$ | $10^{-3}$ |
| | Rank ($r$) | $\{2, 4\}$ |
| | Interval ($\nu$) | $\{50, 100\}$ |
| ZO Fine-Tuner | Batch Size | 16 for LLaMA-1B/8B/Qwen-14B; 4 for OPT-30B |
| | Learning Rate | $\{10^{-6}, 10^{-7}, 10^{-8}\}$ (plus $3 \times 10^{-7}$ for WSC only) |
| | $\epsilon$ | $10^{-3}$ |

reduces to single-dataset training as a special case. We find that training on a single dataset can yield competitive performance with reduced cost. In our experiments, the optimizer is trained on COPA. A comparison between single-dataset and multi-dataset training results is provided in Section D.5. We also block certain gradient flows to reduce memory consumption during learning-to-learn. Specifically, recall that

$$\hat{g}(\theta_t; \omega) = \frac{\mathcal{L}(\theta_t + \epsilon u_t) - \mathcal{L}(\theta_t - \epsilon u_t)}{2\epsilon} u_t$$
$$u_t = \text{PertNN}(\theta_t, \Sigma_{t-1}, \ell_{t-1}; \omega) z_t, \quad z_t \sim \mathcal{N}(0, I_d).$$

The gradient of the ZO loss, defined as $\mathcal{L}_{\text{ZO}}(\theta; \omega) := \mathcal{L}(\theta - \eta \hat{g}(\theta; \omega))$, propagates first to $\hat{g}(\theta_t; \omega)$, and then further through both components used to construct it: the perturbation direction $u_t$ and the finite-difference estimator $\frac{\mathcal{L}(\theta_t + \epsilon u_t) - \mathcal{L}(\theta_t - \epsilon u_t)}{2\epsilon}$. To save memory, we cut off the gradient flow through the finite-difference term, which eliminates the need to back-propagate through the inner loss evaluations and store their activations. Despite this approximation, we still observe strong empirical performance.

# D    ADDITIONAL EXPERIMENTAL RESULTS

## D.1    ADDITIONAL RESULTS ON LEARNING RATE SENSITIVITY

We previously presented the sensitivity of different optimization methods to the learning rate in Section 4.2. Due to space constraints, only a subset of the results was shown. Here, we provide the complete loss curves across the three benchmarks SQuAD, SST-2, and COPA using LLaMA-1B, LLaMA-8B, Qwen-14B, and OPT-30B, as shown in Figure 4 and Figure 5.

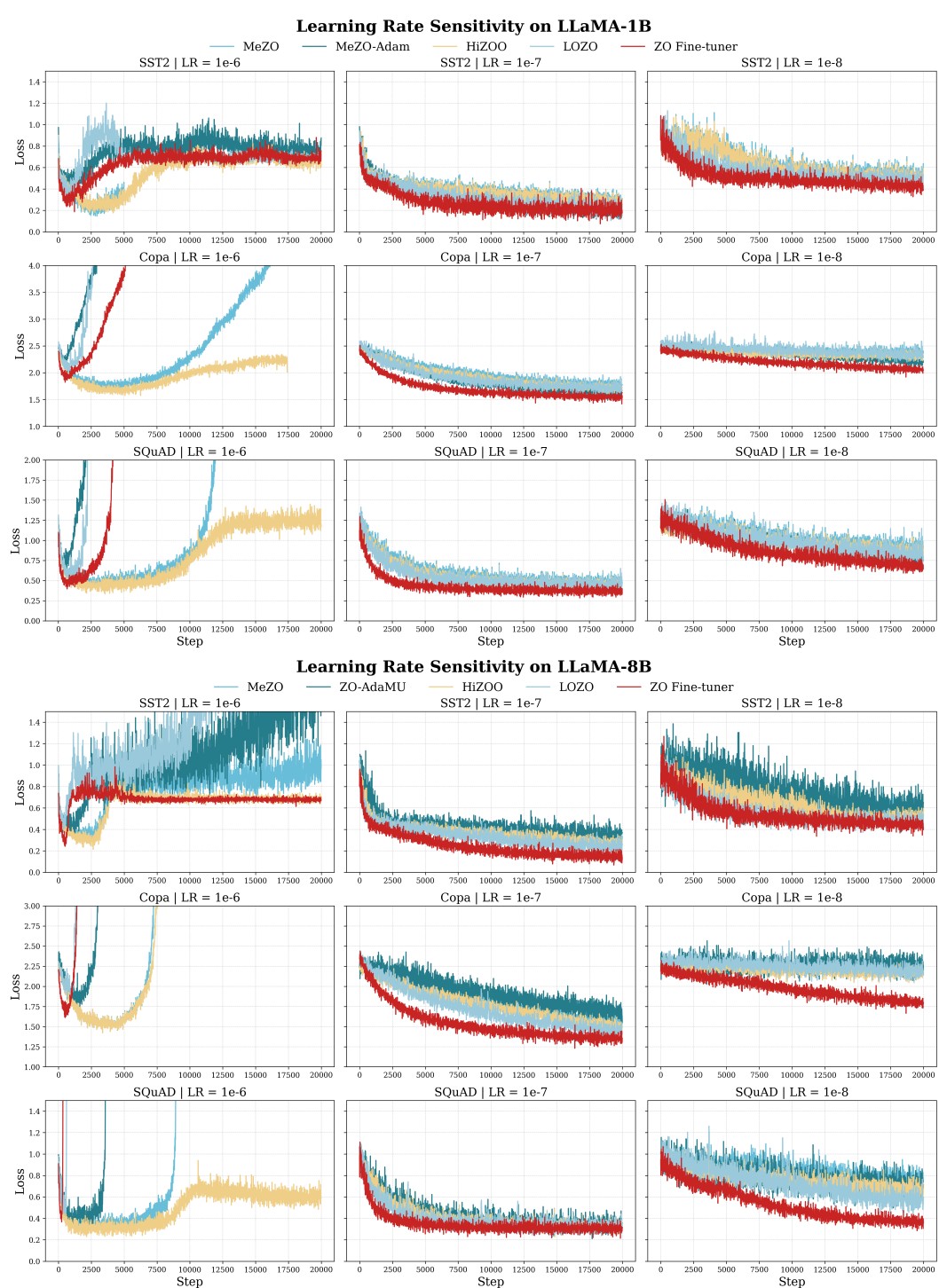

Figure 4: Loss curves under varying learning rates for different optimizers with LLaMA-1B (top) and Qwen-14B (bottom). We report results on SST2, Copa, and SQuAD. For MeZO-Adam, note that the actual learning rates used were $10^{-4}$, $10^{-5}$, and $10^{-6}$, corresponding to the plotted values of $10^{-6}$, $10^{-7}$, and $10^{-8}$, respectively.

Across the grid search over learning rates $10^{-6}$, $10^{-7}$, and $10^{-8}$, the ZO Fine-tuner consistently achieves superior results compared to all baselines when comparing their best-performing settings. On LLaMA-1B, LLaMA-8B, Qwen-14B and OPT-30B, our method exhibits faster convergence and

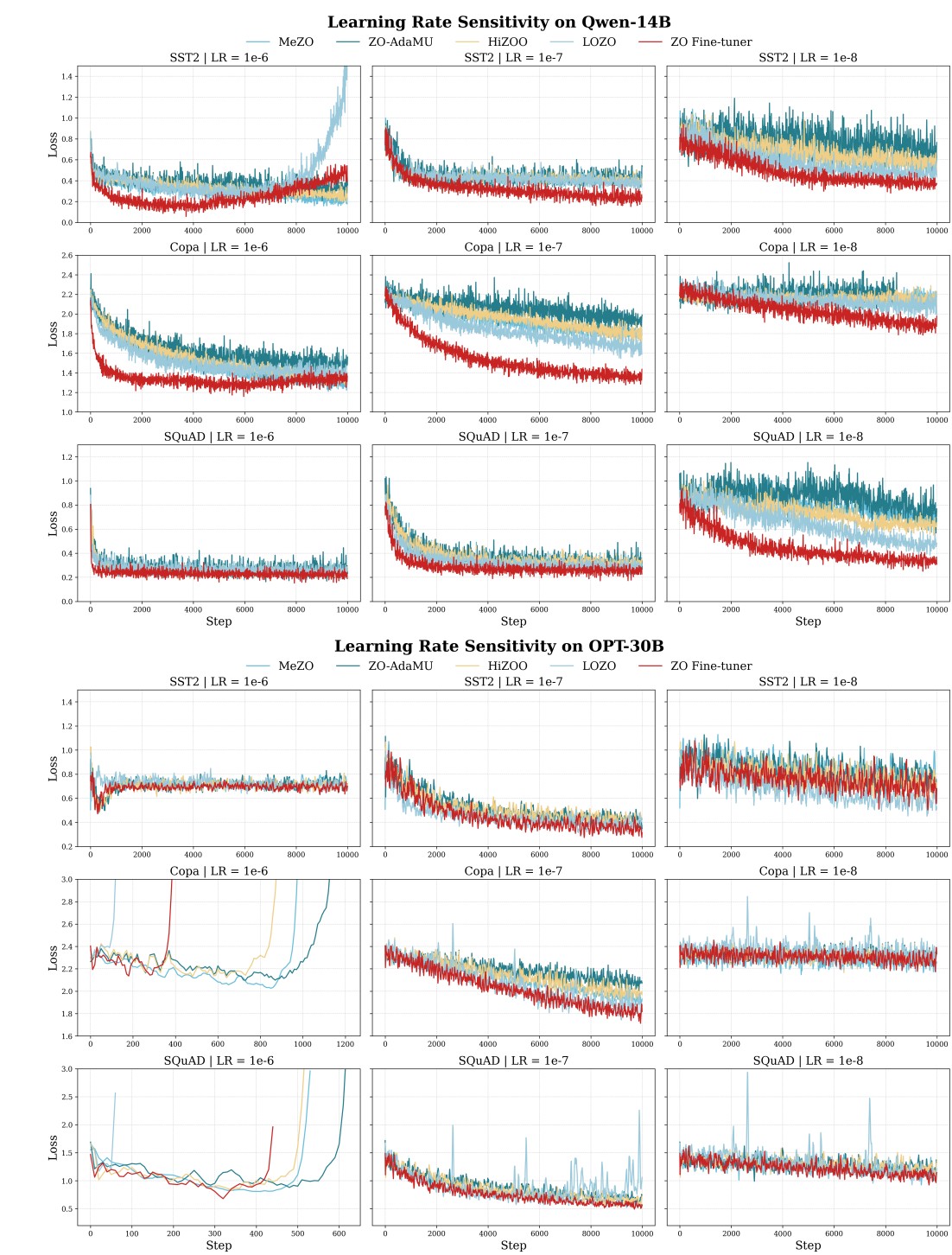

Figure 5: Loss curves under varying learning rates for different optimizers with Qwen-14B (top) and OPT-30B (bottom). We report results on SST2, Copa, and SQuAD.

achieves lower final loss, particularly under the two learning rates $10^{-7}$ and $10^{-8}$. Notably, ZO Fine-tuner often matches or exceeds the best performance of other methods at $10^{-7}$, even when operating at $10^{-8}$ on LLaMA-1B, LLaMA-8B and Qwen-14B.

Table 6: Component-wise runtime breakdown (in seconds and percentage of total time) for different models. All results are tested on DROP and L40S GPU with a batch size of 16 using FP16.

| Model | Generate Var | Perturb Param | Update Param | Compute Loss | Total Time |
|---|---|---|---|---|---|
| LLaMA-1B | 0.025s (3.39%) | 0.052s (7.07%) | 0.021s (2.86%) | 0.631s (86.65%) | 0.729s |
| LLaMA-8B | 0.052s (1.66%) | 0.460s (14.67%) | 0.192s (6.11%) | 2.433s (77.55%) | 3.137s |
| Qwen-14B | 0.119s (2.06%) | 0.395s (6.82%) | 0.164s (2.84%) | 5.106s (88.27%) | 5.785s |
| OPT-30B | 0.142s (1.64%) | 0.214s (2.48%) | 0.090s (1.04%) | 8.183s (94.82%) | 8.630s |

When increasing the learning rate from $10^{-8}$ to $10^{-7}$, ZO Fine-Tuner continues to improve. In contrast, baseline methods tend to suffer from instability at higher learning rates like $10^{-6}$ when increasing from $10^{-7}$, especially on LLaMA-1B and LLaMA-8B. At the low end ($10^{-8}$), many baselines exhibit stagnation, which means their loss decreases slowly or plateaus. This suggests limited adaptivity in low-gradient regimes. These observations underscore the robustness of ZO Fine-Tuner across a wide range of learning rates and tasks, highlighting its strong default behavior even without fine-tuned hyperparameters.

## D.2 TIME ANALYSIS

We further break down the runtime of each component involved in the optimizer and summarize the results in Table 6. Among these, the variance generation step is extremely lightweight. It only accounts for 3.39% of total runtime on LLaMA-1B, and less than 2.06% on larger models such as LLaMA-8B, Qwen-14B, and OPT-30B. This highlights the efficiency of our design: although we introduce an additional learned component to control perturbation variance, it imposes almost no computational overhead.

This can be easily explained as we only employ a lightweight neural network for each parameter block. More specifically, a two-layer MLP with just 32 hidden units. In addition, both the input and output of these networks are compressed, further reducing the computational cost.

In contrast, the dominant cost comes from loss computation, which includes forward passes for both positive and negative perturbations. This accounts for over 77%–95% of total runtime and is intrinsic to all zeroth-order optimization frameworks. Overall, our method introduces minimal additional cost while achieving adaptive and effective optimization.

## D.3 MEMORY ANALYSIS

Table 7 reports the peak GPU memory usage of various optimization methods across different model sizes on the SST-2 dataset. We observe that all zeroth-order (ZO) methods, including MeZO, LOZO, and HiZOO, exhibit similar memory footprints. The only notable exception is ZO-AdaMU, which incurs higher memory usage due to its additional momentum tracking. Compared to the first-order method like Adam, all ZO methods consume significantly less memory, highlighting the efficiency of ZO-based optimization. Notably, our ZO Fine-Tuner achieves comparable memory usage to other ZO baselines, indicating that it introduces no additional memory overhead beyond standard ZO designs.

Table 7: Peak GPU memory usage (GB) of different optimization methods across models on the SST-2 dataset, using batch size = 1 and FP16 precision.

| Method | LLaMA-1B | LLaMA-8B | Qwen-14B | OPT-30B |
|---|---|---|---|---|
| MeZO | 5 | 20 | 35 | 61 |
| LOZO | 5 | 20 | 35 | 61 |
| HiZOO | 6 | 23 | 40 | 65 |
| ZO-AdaMU | 9 | 39 | 69 | 122 |
| ZO Fine-Tuner | 5 | 21 | 36 | 62 |
| FO-SGD | 9 | 40 | 74 | 126 |
| FO-Adam | 13 | 84 | 163 | 316 |

Table 8: Time and memory cost of meta-training the ZO Fine-Tuner in our L2L framework. GPU memory usage and GPU time (in minutes) are reported for different foundation models. This cost is incurred only once per base model, and the trained fine-tuner can be reused across downstream tasks.

| Model | GPU Memory (GB) | Meta-training GPU Time (min) |
|---|---|---|
| LLaMA-1B | 13 | 3 |
| LLaMA-8B | 83 | 15 |
| Qwen-14B | 150 | 25 |
| OPT-30B | 332 | 51 |

### D.4 COST OF LEARNING TO LEARN

We also assess the time and memory overhead incurred during the training of the ZO Fine-Tuner in table 8. In general, L2L takes approximately 2.4× the time of standard first-order fine-tuning. This is expected, as the L2L process inherently includes a full fine-tuning phase using SGD. However, importantly, this cost is incurred only **once**, as a single ZO Fine-Tuner trained for a given model can be reused across diverse downstream tasks, effectively amortizing the training cost.

### D.5 COMPARISON BETWEEN SINGLE-DATASET AND MULTI-DATASET TRAINING

We also compare the performance of ZO Fine-tuner under single-dataset and multi-dataset training settings. In the multi-dataset setting, we construct a diverse training set by selecting one representative dataset from each task type: SST-2 for sentiment analysis, CB and COPA for natural language inference, and SQuAD for question answering. For the single-dataset setting, the optimizer is trained solely on COPA.

The multi-dataset setting could lead to better performance. However, as shown in Figure 6, in some cases, the ZO Fine-tuner trained on a single dataset can outperform its multi-dataset counterpart. Overall, the two settings yield comparable performance. Single-dataset training is also simpler to implement and tune, while still achieving competitive results. And that's why we choose to use it throughout the main experiments.

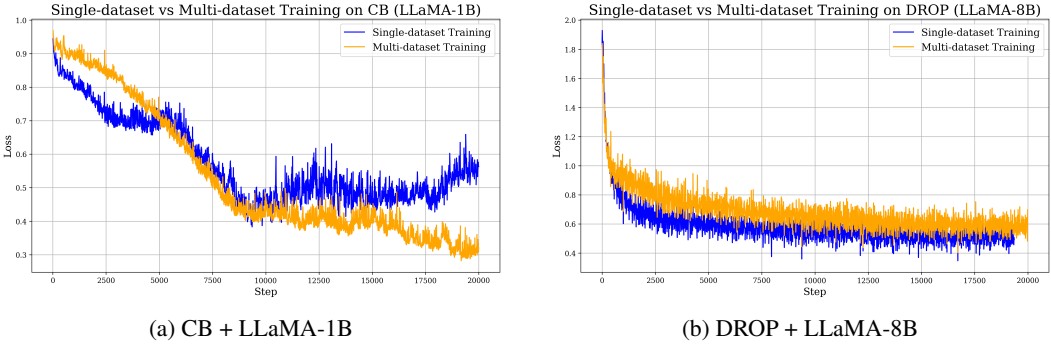

(a) CB + LLaMA-1B          (b) DROP + LLaMA-8B

Figure 6: Comparison of inference loss between ZO Fine-tuners trained with single-dataset and multi-dataset settings. Results are reported on CB task using LLaMA-1B (left) and on DROP task LLaMA-8B (right). The single-dataset variant is trained solely on COPA, while the multi-dataset variant is jointly trained on COPA, SST-2, and SQuAD.

## E THEORETICAL ANALYSIS

In this section, we formally discuss our theoretical results and derive theorem 1. First, we set up the notations and definition we need and formally present the theorem.

**Definition 1** (Expected Loss Change). *The expected change in loss after performing a one-step update from parameter $\theta_t$ is defined as*

$$d(\theta_t) := \mathbb{E}[\mathcal{L}(\theta_{t+1}) \mid \theta_t] - \mathcal{L}(\theta_t).$$

**Assumption 1** (Local $r$-effective rank). *Let $G(\theta_t) = \max_{(x,y) \in D} \|\nabla \mathcal{L}(\theta_t; \{(x,y)\})\|$. There exists a matrix $H(\theta_t) \leq \ell \cdot I_d$ such that:*

1. *For all $\theta$ such that $\|\theta - \theta_t\| \leq \eta d G(\theta_t)$, we have $\nabla^2 \mathcal{L}(\theta) \preceq H(\theta_t)$.*
2. *The effective rank of $H(\theta_t)$, i.e. $\operatorname{tr}(H(\theta_t))/\|H(\theta_t)\|_{\mathrm{op}}$, is at most $r$.*

**Theorem 2.** *Under Assumption 1, the expected loss change after one-step update of MeZO has upper bound as follows, where $\Sigma_{MB} = \operatorname{Cov}(\nabla \mathcal{L}(\theta_t; \{(x_i, y_i)\}))$:*

$$d_{\mathrm{MeZO}}(\theta_t) = \mathbb{E}[\mathcal{L}(\theta_{t+1})|\theta_t] - \mathcal{L}(\theta_t)$$

$$\leq -\eta \|\nabla \mathcal{L}(\theta_t)\|^2 + \frac{\eta^2 \ell}{2} \cdot \left( \frac{dr + d - 2}{d + 2} + 1 \right) \cdot \left( \|\nabla \mathcal{L}(\theta_t)\|^2 + \frac{1}{B} \operatorname{tr}(\Sigma_{MB}(\theta_t)) \right)$$

**Assumption 2** (Local Block-wise $r_i$-Effective Rank). *The Hessian matrix $H(\theta_t)$ in Assumption 1 satisfies the following property: $H(\theta_t) = \operatorname{diag}(H_1(\theta_t), \ldots, H_m(\theta_t))$ and $r_i := \operatorname{tr}(H_i(\theta_t))/\|H_i(\theta_t)\|_{\mathrm{op}}$ have different upper bounds $r_i$.*

**Theorem 3.** *Under Assumption 1 and Assumption 2 and ideal situation, assigning distinct perturbation variances across parameter blocks can yield a tighter upper bound than that of $d_{\mathrm{MeZO}}(\theta_t)$.*

Assumption 1 and Theorem 1 are directly from the theoretical analysis of MeZO Malladi et al. (2023). MeZO states the Lipschitz condition alone does not guarantee convergence in high-dimensional settings and it is necessary to **leverage the low-rank structure of the Hessian matrix**. Assumption 2 is consistent with the actual situation, which has been deeply researched and checked by work like Adam-mini (Zhang et al., 2024b;a).

In this section, we consider a setting where, at each iteration, we sample a perturbation block-by-block: for the $i$-th parameter block, we draw noise from the distribution $N(0, \sigma_i I_{d_i})$, apply the perturbation to the $i$-th block of parameters, and perform a zeroth-order update accordingly. Let the total number of blocks be $b$, and denote by $\mathcal{L}(\theta_{t,j})$ the loss after perturbing the $j$-th block at iteration $t$. We further denote the full loss after perturbing all $b$ blocks as $\mathcal{L}(\theta_{t+1})$, and let $\nabla_j \mathcal{L}(\theta_{t,j})$ denote the $j$-th block of the gradient evaluated at $\theta_{t,j}$. Here, we adopt the sphere (normalized-Gaussian) perturbation used in the original MeZO analysis for its built-in step-size control. An analogous convergence form also holds for Gaussian perturbations, as shown in prior work(Malladi et al., 2023), when the probability of large updates $\|\theta_{t+1} - \theta_t\|$ is kept small, which ensures the required local assumptions hold with high probability.

*Proof.* As shown in Theorem 2, the expected loss decrease under MeZO is bounded by

$$d_{\mathrm{MeZO}}(\theta_t) = \mathbb{E}[\mathcal{L}(\theta_{t+1})|\theta_t] - \mathcal{L}(\theta_t)$$

$$\leq -\eta \|\nabla \mathcal{L}(\theta_t)\|^2 + \frac{\eta^2 \ell}{2} \cdot \left( \frac{dr + d - 2}{d + 2} + 1 \right) \cdot \left( \|\nabla \mathcal{L}(\theta_t)\|^2 + \frac{1}{B} \operatorname{tr}(\Sigma_{MB}(\theta_t)) \right)$$

Since each block-wise gradient estimate is still an unbiased estimator of the true gradient restricted to the corresponding block, we can get:

$$\mathbb{E}[\mathcal{L}(\theta_{t,j+1})|\theta_{t,j}] - \mathcal{L}(\theta_{t,j}) \leq$$

$$- \eta \sigma_j^2 \|\nabla_j \mathcal{L}(\theta_{t,j})\|^2 + \frac{\eta^2 \sigma_j^4 \ell}{2} \cdot \left( \frac{dr_j + d - 2}{d + 2} + 1 \right) \cdot \left( \|\nabla_j \mathcal{L}(\theta_{t,j})\|^2 + \frac{1}{B} \operatorname{tr}(\Sigma_{MB,j}(\theta_{t,j})) \right)$$

By summing both sides over $j = 1$ to $b$ and taking expectation, we eliminate the dependence on $\theta_{t,j}$: the right-hand side becomes a function of $\theta_t$ only, while the left-hand side depends only on $\theta_{t+1}$ and $\theta_t$. This yields:

$$\mathbb{E}[\mathcal{L}(\theta_{t+1})|\theta_t] - \mathcal{L}(\theta_t) \leq -\eta \sum_{j=1}^{b} \sigma_j^2 \mathbb{E}[\|\nabla_j \mathcal{L}(\theta_{t,j})\|^2 | \theta_t]$$

$$+ \sum_{j=1}^{b} \frac{\eta^2 \sigma_j^4 \ell}{2} \cdot \left( \frac{dr_j + d - 2}{d + 2} + 1 \right) \cdot \left( \mathbb{E}[\|\nabla_j \mathcal{L}(\theta_{t,j})\|^2 | \theta_t] + \frac{1}{B} \operatorname{tr}(\mathbb{E}[\Sigma_{MB,j}(\theta_{t,j})|\theta_t]) \right)$$

$$= \sum_{j=1}^{b} \left[ -\eta \sigma_j^2 \|\nabla_j \mathcal{L}(\theta_t)\|^2 + \frac{\eta^2 \sigma_j^4 \ell}{2} \cdot \left( \frac{dr_j + d - 2}{d + 2} + 1 \right) \cdot \left( \|\nabla_j \mathcal{L}(\theta_t)\|^2 + \frac{1}{B} \operatorname{tr}(\Sigma_{MB}(\theta_t)) \right) \right]$$

The equality in the last line follows from the condition that the Hessian matrix is block-diagonal according to Assumption 2. Specifically, when updating block $j_1$, the change in the gradient of block $j_2$ ($j_2 \neq j_1$) can be expressed as:

$$\nabla_{j_2} \mathcal{L}(\theta_{t,j_1}) - \nabla_{j_2} \mathcal{L}(\theta_t) = \int_0^1 H_{j_2 j_1}(\theta_t + sP_{j_1}\delta)\,\delta\,ds,$$

where $P_{j_1}$ denotes the projection onto block $j_1$, and $H_{j_2,j_1}(\cdot)$ is the $(j_2, j_1)$ block of the Hessian. The perturbation direction $\delta$ is sampled from the standard multivariate normal distribution and scaled by the corresponding block-wise variance, i.e., $\delta \sim \mathcal{N}(0, \sigma_{j_1}^2 I_{d_{j_1}})$ for block $j_1$. Under the block-diagonal assumption, $H_{j_2,j_1}(\cdot) = 0$ for all $j_2 \neq j_1$, and thus cross-block gradient changes vanish.

Even if Assumption 2 does not hold exactly, the effect of cross-block interactions can still be controlled by bounding the operator norm of the off-diagonal blocks of the Hessian. As long as these terms remain small, the overall error introduced in the bound remains negligible.

Note that if we set $\sigma_j = 1$ for all $j$, our upper bound reduces to the standard MeZO bound:

$$\mathbb{E}[\mathcal{L}(\theta_{t+1})|\theta_t] - \mathcal{L}(\theta_t)$$

$$\leq \sum_{j=1}^b \left[ -\eta \|\nabla_j \mathcal{L}(\theta_t)\|^2 + \frac{\eta^2 \ell}{2} \cdot \left( \frac{dr_j + d - 2}{d+2} + 1 \right) \cdot \left( \|\nabla_j \mathcal{L}(\theta_t)\|^2 + \frac{1}{B} \mathrm{tr}(\Sigma_{MB}(\theta_t)) \right) \right]$$

$$\leq -\eta \|\nabla \mathcal{L}(\theta_t)\|^2 + \frac{\eta^2 \ell}{2} \cdot \left( \frac{dr + d - 2}{d+2} + 1 \right) \cdot \left( \|\nabla \mathcal{L}(\theta_t)\|^2 + \frac{1}{B} \mathrm{tr}(\Sigma_{MB}(\theta_t)) \right)$$

where $r$ is the (uniform) effective rank used in MeZO and $r \geq r_j$ for any $j$. Therefore, by optimizing $\sigma_j$ for each block according to its local structure (e.g., $r_j$), we can obtain a strictly tighter upper bound than $d_{\mathrm{MeZO}}(\theta_t)$. $\qquad\square$

