# OpenReview forum: "Learning a Zeroth-Order Optimizer for Fine-Tuning LLMs"
_ICLR.cc/2026/Conference — Submitted to ICLR 2026_

### Official Review · Reviewer_rsyj · 2025-11-01

**Soundness:** 2
**Presentation:** 2
**Contribution:** 2
**Rating:** 4
**Confidence:** 3

**Summary:**

This paper introduces a learned zeroth-order (ZO) optimizer for memory-efficient LLM fine-tuning. Instead of fixed perturbation scales (e.g., MeZO), the method learns block-level variance policies through lightweight neural modules and normalizes their overall scale. The optimizer is meta-trained once and reused across tasks and models. Experiments show consistent improvements over existing ZO approaches with minimal overhead.

**Strengths:**

1. Block‑wise variance learned by tiny MLPs; total aux parameters <2 MB even for OPT‑30B

2. A finetuner learned on COPA transfers across 7 datasets and to LLaMA‑3.1‑8B‑Instruct, retaining gains。

3. Near‑inference memory and negligible extra compute (e.g., OPT‑30B 62 GB vs 316 GB for FO‑Adam

**Weaknesses:**

1. According to Eq. 4, with anisotropic Gaussians E[uₜuₜ^⊤]=Σₜ, so the expected update is Σₜ∇L, not ∇L. The method neither whitens by Σₜ^{-1} nor analyzes the induced bias; fixing ‖Σₜ‖_F cannot remove directional preconditioning.

2. The optimizer is trained to minimize one‑step post‑update loss (Eq. 5), which risks greedy behavior and gives no guarantees for long‑horizon stability under accumulated ZO noise.

3. In Algorithm 2, after updating PertNN (line 8), they advance the model with SGD to produce the next training state; at deployment there is no backprop at all. This teacher‑forcing bias is not addressed.

4. Per block, inputs are only Mean, Var, previous σ, and two losses. These statistics are loosely related to local curvature/gradient structure, undermining the “block‑diagonal Hessian” motivation.

5. The method is compared only to ZO baselines. Including parameter-efficient first-order baselines at similar memory budgets (e.g., LoRA/QLoRA with 8-bit optimizer states) would clarify the method’s place in the practical landscape.

6. The informal theorem hinge on approximate block‑diagonality, yet the paper reports no quantitative measure of off‑diagonal energy/spectral norms on real LLMs.

**Questions:**

The method is motivated by the assumption that curvature is approximately block-structured. Have you examined Hessian or Fisher block interactions in practice?

How does the approach compare to well-tuned LoRA/QLoRA baselines under similar memory constraints?

---

> ### Author Response · Authors · 2025-12-03
>
> We appreciate the reviewer’s feedback. Below, we address the concerns through detailed clarifications of the scope and design choices of our work.
>
> > **W1: Directional preconditioning**
>
> We would like to clarify that completely eliminating directional preconditioning is not our goal. The purpose of the normalization is to **remove the impact of $\Sigma$ on the global learning rate**, thereby ensuring a fair comparison under a fixed learning rate. The directional preconditioning does not cause such an issue, and we see no need for normalizing that factor.
>
> > **W2 & W3 & W4: Additional biases**
>
> We acknowledge that the biases mentioned by the reviewer may indeed exist in our current methodology, and that the inputs to PertNN are not sufficient to fully recover the curvature or Hessian structure. However, our design must balance practical feasibility (memory constraints, training cost) with performance. Given the strong empirical results across multiple models and datasets, we believe that these potential biases do not undermine the practical usefulness of our method.
>
> > **W5 & Q2: Comparison with LoRA/QLoRA**
>
> We would like to clarify that our goal in this paper is to improve the optimization efficiency of ZO methods in the full-parameter fine-tuning regime, rather than to compete with the most aggressively engineered memory-saving schemes. We agree that a well-tuned LoRA/QLoRA baseline can achieve strong performance with memory usage comparable to, or even lower than, full-parameter ZO methods. However, we view this as a general limitation of the full-parameter ZO setting, not a weakness specific to our approach.
>
> Importantly, **ZO Fine-tuner is orthogonal to such techniques**: the same L2L perturbation network could, in principle, be **combined with quantization, sparsity, or LoRA-style updates to further reduce memory**. Our current work is a first step that demonstrates the promise of L2L for ZO optimization on LLMs, and we hope it will **motivate future extensions** that incorporate these mechanisms to achieve even better memory–performance trade-offs.
>
> >**W6 & Q1: Quantifying the block-diagonalness of Hessians**
>
> Unfortunately, computing and analyzing full Hessians for LLMs is very expensive and technically challenging. For this reason, we rely on prior work that empirically observes approximately block-structured Hessians in standard Transformer architectures [1,2,3]. This observation serves only as **high-level motivation for our blockwise design**, and the **practicality and usefulness of our approach are ultimately supported by its empirical performance.** We agree with the reviewer that this point deserves to be investigated further by considering Hessian or Fisher block interaction, and we view this an interesting future work.
>
> [1] Zhang, Yushun, et al. "Adam-mini: Use fewer learning rates to gain more." arXiv preprint arXiv:2406.16793 (2024).\
> [2] Zhang, Yushun, et al. "Why transformers need adam: A hessian perspective." Advances in Neural Information Processing Systems 37 (2024): 131786–131823.\
> [3] Ormaniec, Weronika, Felix Dangel, and Sidak Pal Singh. "What does it mean to be a transformer? insights from a theoretical hessian analysis." arXiv preprint arXiv:2410.10986 (2024).

---

### Official Review · Reviewer_bS2A · 2025-11-02

**Soundness:** 3
**Presentation:** 3
**Contribution:** 3
**Rating:** 2
**Confidence:** 5

**Summary:**

This paper proposes a new zeroth-order optimizer for LLM fine-tuning that learns a variance term for perturbation at each layer for an LLM. These learned perturbation strategies then generalize to other fine-tuning tasks that use the same foundation model. The ZO Fine-Tuner approach outperforms existing methods such as MEZO on training loss.

**Strengths:**

The paper is fairly clearly presented, and the main idea is sensible. The main hypothesis, that perturbation strategies can generalize from task to task with the foundation model fixed, is interesting and does seem to be validated appropriately by the results in Table 3. The memory usage and runtime measurements in Section 4.3 do a good job of giving us a sense of the cost of the method.

**Weaknesses:**

As far as I can tell, there are just no validation/test measurements for anything done in this paper, so we have no idea how ZO Fine Tuner might affect generalization. Even though the paper is otherwise strong, the lack of any (unless I missed something!) validation/test measurements seems fatal.

**Questions:**

Where (I hope I just missed it) are the validation/test measurements in the paper to show generalization or applicability to downstream tasks on data different from that on which the model was trained? What would Table 1 look like if the numbers reported were on the test set rather than the training set?

---

> ### Author Response · Authors · 2025-12-03
>
> We thank the reviewer for recognizing the strengths of our paper. We believe **there are significant misunderstandings regarding our experimental setup**, which we clarify below.
>
> For all experiments reported in Table 1, the ZO Fine-tuner is **trained only once on COPA**, as stated in lines 364–365. After this training phase, we freeze the ZO Fine-tuner and apply it directly to the other datasets (SST-2, CB, SQuAD, WSC, BoolQ, DROP) without any additional training or adaptation. Therefore, in Table 1, **all results except those in the two COPA columns are computed on test data that the ZO Fine-tuner has never seen during training**. These results already evaluate the generalization and transferability of the ZO Fine-tuner to unseen data and to unseen tasks.

---

### Official Review · Reviewer_TZJo · 2025-11-03

**Soundness:** 3
**Presentation:** 3
**Contribution:** 3
**Rating:** 2
**Confidence:** 4

**Summary:**

The paper introduces ZO Fine-tuner, a learning-to-learn zeroth-order optimizer for LLM fine-tuning. Instead of drawing perturbations from a fixed distribution, it attaches a tiny auxiliary “PertNN” to each parameter block (e.g., Q/K/V, projections, embeddings) to predict that block’s perturbation variance. A simple norm-budget normalization keeps variance scale decoupled from the step size. The method is motivated by a block-diagonal Hessian view and an informal result suggesting per-block adaptive variance can tighten the expected one-step loss bound versus standard MeZO. Training uses a short first-order meta-learning phase done once per base model; after that, the learned optimizer is reused across tasks and derivative checkpoints of the same model family. Experiments on 4 LLMs across 7 datasets (28 settings) show lower converged loss in most cases and an average accuracy gain of about 2.5% over ZO baselines, with negligible runtime and memory overhead relative to MeZO. A small transfer test indicates the optimizer trained on LLaMA-8B also works on LLaMA-8B-Instruct. Ablations highlight the benefits of the variance normalization trick, periodic resets during meta-training, and block-wise sharing over layer-wise sharing.

**Strengths:**

Originality: Framing ZO fine-tuning as learning a per-block perturbation variance via tiny auxiliary networks is a neat twist that removes hand-crafted noise schedules and makes ZO more adaptive. The norm-budget trick to decouple step size from variance scale is simple but fresh and broadly applicable.

Quality: The method is implemented cleanly with low overhead, and the experiments cover multiple LLM sizes and seven benchmarks. Useful ablations (normalization, reset strategy, sharing granularity) help isolate which pieces matter. Runtime/memory tables demonstrate practical deployability.

**Weaknesses:**

1. The “cross-model generalization” evidence only tests transfer from LLaMA-8B to LLaMA-8B-Instruct, which share the same architecture and most parameter shapes. This is effectively within the same model family, not true cross-model generalization. The claim of “train once, reuse widely” is therefore overstated.

2. The experiments mostly use short-sequence or classification-style tasks (SST-2, CB, COPA, BoolQ, etc.). There is no evidence that the proposed optimizer generalizes to long-sequence dataset.

**Questions:**

1.Generalization beyond text classification – Have you evaluated ZO Fine-tuner on longer-sequence.

2.Good analysis on Training loss and Inference loss in experment, does author try to analyze the accuracy curve in experment?

---

> ### Author Response · Authors · 2025-12-03
>
> We thank the reviewer for the detailed comments. We have addressed all concerns through additional experiments (more datasets, accuracy curves) and further clarification of our experimental settings. Below we provide a detailed response to each of the reviewer's concern.
>
> > **W1: Transferability**
>
> Our intention was **not to claim transferability across different model families** (e.g., from LLaMA to Qwen), but rather to demonstrate that a ZO Fine-tuner **learned on a base checkpoint can be reused for derivative checkpoints within the same family**. In particular, our experiment shows that a ZO Fine-tuner meta-trained on LLaMA-3.1-8B can be directly reused to fine-tune LLaMA-3.1-8B-Instruct, an already fine-tuned derivative of the base model. We believe this provides strong evidence of the method’s practical usefulness, as a recent LLM supply-chain study suggests that **many HuggingFace models are derivatives of a small number of base checkpoints (notably Qwen and LLaMA) [1]**. In such a setting, a ZO Fine-tuner released for a base model (e.g., by the Qwen or LLaMA team) could, in principle, be reused across a wide range of derived models within that family, without retraining the ZO Fine-tuner from scratch.
>
> > **W2 & Q1: More datasets**
>
> While many of the benchmarks in Table 1 are classification tasks, **our evaluation is not restricted to short-sequence classification**. In particular, **SQuAD and DROP are generation tasks with significantly longer contexts**. Moreover, our dataset choice (COPA, SST-2, CB, SQuAD, WSC, BoolQ, DROP) closely **follows the setups used in recent ZO papers for LLMs [2,3,4,5]**, which also evaluate several classification tasks plus SQuAD/DROP-type reading comprehension. This allows direct comparison within the prevailing ZO evaluation regime. To respond to the request of adding more datasets, we have **additionally evaluated our approach compared to MeZO on two more datasets**, ReCoRD (longer context) and WIC, on LlaMA3.1-8B, and the results are shown below:
>
> `Llama3.1-8B+ReCoRD`
> | **Setting**   | **Avg Loss \[0,4k\)**     | **Avg Loss \[4k,8k\)**    | **Avg Loss \[8k,12k\)**   | **Avg Loss \[12k,16k\)**  | **Avg Loss \[16k,20k\)**  | **Final Accuracy** |
> |:---|:----:|:---:|:---:|:---:|:----:|:---:|
> | MeZO  | 2.5868   | 2.5683  | 2.5443 | 2.5327 | 2.5178 | 0.85 |
> | Ours | 2.5129 | 2.4480 | 2.4217| 2.4093 | 2.4027| 0.85 |
>
> `Llama3.1-8B+WIC`
> | **Setting**  | **Avg Loss \[0,4k\)**     | **Avg Loss \[4k,8k\)**    | **Avg Loss \[8k,12k\)**   | **Avg Loss \[12k,16k\)**  | **Avg Loss \[16k,20k\)**  | **Final Accuracy** |
> |:---|:-----:|:--:|:---:|:---:|:---:|:---:|
> | MeZO | 0.6919 | 0.6858  | 0.6824 | 0.6729 | 0.6703 |  0.55 |
> | Ours  | 0.6918 | 0.6862 | 0.6803 | 0.6683 | 0.6589 | 0.57|
>
> > **Q2: Accuracy Curve**
>
> In this work, we deliberately **focused on loss curves because our main contribution is on the optimization side**. The optimizer’s direct goal is to minimize the loss rather than directly maximize accuracy, and we therefore believe that comparing the loss across baselines is the fairest approach when evaluating optimizers. However, we have now **added example accuracy curves in Appendix D.4** on SST-2 and SQuAD with LLaMA-3.1-8B, comparing our ZO Fine-tuner with MeZO. These plots clearly show that ZO Fine-tuner achieves faster convergence and higher final accuracy than MeZO. A more quantitative comparison is given below:
>
> `SST-2 accuracy curve for LLaMA-3.1-8B`
>
> | Method           | 5k    | 10k   | 15k   | 20k   |
> |------------------|-------|-------|-------|-------|
> | **MeZO**         | 0.8624 | 0.9014 | 0.9151 | 0.9197 |
> | **ZO Fine-tuner**| 0.9037 | 0.9381 | 0.9461 | 0.9518 |
>
> `SQuAD F1 curve for LLaMA-3.1-8B`
> | Method           | 5k    | 10k   | 15k   | 20k   |
> |------------------|-------|-------|-------|-------|
> | **MeZO**         | 0.8202 | 0.8834 | 0.8990 | 0.8952 |
> | **ZO Fine-tuner**| 0.8972 | 0.8975 | 0.9056 | 0.9006 |
>
> [1] Rahman, Mohammad Shahedur, Peng Gao, and Yuede Ji. "Hugginggraph: Understanding the supply chain of llm ecosystem." Proceedings of the 34th ACM International Conference on Information and Knowledge Management. 2025\
> [2] Tan, Qitao, et al. "Harmony in divergence: Towards fast, accurate, and memory-efficient zeroth-order llm fine-tuning." arXiv preprint arXiv:2502.03304 (2025).\
> [3] Zhao, Hanzhen, et al. "PaZO: Preconditioned Accelerated Zeroth-Order Optimization for Fine-Tuning LLMs." The Thirty-ninth Annual Conference on Neural Information Processing Systems.\
> [4] Malladi, Sadhika, et al. "Fine-tuning language models with just forward passes." Advances in Neural Information Processing Systems 36 (2023): 53038-53075.\
> [5] Zhao, Yanjun, et al. "Second-order fine-tuning without pain for llms: A hessian informed zeroth-order optimizer." arXiv preprint arXiv:2402.15173 (2024).

---

### Official Review · Reviewer_7BNG · 2025-11-03

**Soundness:** 3
**Presentation:** 3
**Contribution:** 2
**Rating:** 4
**Confidence:** 4

**Summary:**

This paper proposes ZO Fine-tuner, a learned zeroth-order optimizer for LLM fine-tuning. The authors propose using tiny per-block PertNN networks to predict block-wise perturbation variances, and a normalization step fixes the overall variance budget so a single learning rate remains stable. The fine-tuner is trained once and reused across various downstream tasks. Experiments show lower loss and imporved accuracy across a collection of models/tasks with negligible time and memory overhead.

**Strengths:**

- Meta optimizer is trained once and reused across tasks and model derivatives with consistant gains over MeZO.
- Algorithm is very lightweight and adds minimum memory/time overhead compared to MeZO.
- The paper conducts detailed experimentation across different models and datasets to show generalizability.

**Weaknesses:**

The paper's evaluation is a bit narrow, it only compares against optimizer variants of MeZO (HiZOO/LOZO) and ignores other memory-focused fine-tuning approaches. There exists low-memory baselines such as ZO + quantization workflows, first order + quantization methds (e.g. QLoRA) and recent sparse-ZO approaches (e.g. sparse-MeZO and SenseZOQ)

Appendix D.3 shows OPT-30B requires 62G vram with ZO-Fine-tuner, while prior work demonstrates far better memory efficiency, QLoRA fine-tuned a 65B model on a 48G gpu, and SensZOQ fine-tuned a 7B model within 8G of vram. Since the authors emphasize memory efficiency, comparisons against these stronger, memory focused baselines is important but currently missing.

1. QLoRA https://arxiv.org/abs/2305.14314
2. Sparse MeZO https://arxiv.org/abs/2402.15751
3. SenseZOQ https://openreview.net/pdf?id=myYzr50xBh

**Questions:**

1. How does ZO-Fine-tuner perform in terms of memory usage and convergence speed when compared against
  - Sparse MeZO with its dynamic mask implementation
  - SenseZOQ which uses a static 0.1% weight mask + 4bit quantization
  - QLoRA, which is the first order + quantization baseline

2. It is known from the literature that ZO methods suffers from slow convergence speed issues, so how does ZO-Fine-tuner's convergence speed compare with ZeRO-3 offloading, since both trade training speed for memory reduction? What is the accuracy/performance gap between ZO-Fine-tuner and full FO fine-tuning under comparable resource settings?

3. Table 3 reports no FO baseline for direct comparison, could the authors include results for FO-Adam or FO-SGD to quantify how much performance is lost when moving from first-order to zeroth-order optimization?

---

> ### Author Response · Authors · 2025-12-03
>
> We thank the reviewer for the instructive feedback. We believe the reviewer’s concerns **mostly arise from a misconception of our work’s scope**. Below, we address these concerns through additional experiments (FO baselines) and a more detailed clarification of the scope of our work.
>
> > **W1 & Q1 & Q2: Comparison with other memory-efficient fine-tuning approaches (QLoRA, Sparse MeZo, SenseZOQ, ZeRO-3 offloading).**
>
> We agree that these baselines are valuable and represent interesting directions for future work. However, **we view the suggested directions as largely orthogonal to the current focus of this paper.**
>
> Our goal in this work is not to introduce a new technique for further reducing the memory footprint of existing ZO methods, but rather to demonstrate **the potential of learning-to-learn (L2L) methods for improving optimization quality within the standard MeZO-style ZO setting, with minimal hand-crafted design.** Accordingly, all our baselines (MeZO, HiZOO, MeZO-Adam) are evaluated under the same regime, where we fine-tune all LLM parameters in FP16, without quantization or any other specialized structures.
>
> **We therefore believe that direct comparisons to the reviewer’s suggested baselines (QLoRA, Sparse MeZO, SenseZOQ, ZeRO-3 offloading, etc.) would not be directly aligned with the present scope**, as these methods rely on specialized designs such as quantization, offloading, or sparsity. It is indeed possible for well-designed FO/ZO methods to achieve lower memory usage and even superior performance compared to plain ZO methods with full-parameter fine-tuning like ours, or to further reduce memory on top of traditional ZO regimes. However, similar levels of specialized design (e.g., CPU offloading, sparse or quantized ZO, LoRA-style updates) could also be applied on top of the ZO Fine-tuner. Designing and thoroughly evaluating such variants is an important but separate direction, and is beyond the scope of this work.
>
> > **Q3: Comparison with FO baseline.**
>
> We now report the final accuracy across all datasets for LLaMA-3.2-1B when fine-tuned with Adam, as well as additional FO results on SST-2 with multiple models. As expected, **FO methods generally outperform ZO methods,** while our method remains competitive with only a **modest performance gap**.
>
> `Llama3.2‑1B`
> | **Setting** | **Copa** | **CB** | **SQuAD**| **WSC**| **BoolQ**| **DROP**|
> |:-----|:---:|:----:|:----:|:----:|:----:|:---:|
> | MeZO   | 0.75 | 0.70  | 0.75 | 0.62 | 0.63 | 0.29 |
> | Ours | 0.80 | 0.73  | 0.80 | 0.57 | 0.66 | 0.32 |
> | FO  | 0.76 | 0.75  | 0.82 | 0.60 | 0.69 | 0.45 |
>
> `SST2 dataset`
> | **Setting** | **Llama3.2‑1B** | **Llama3.1‑8B** | **Qwen2.5‑14B** |
> |:----:|:----:|:---:|:----:|
> | MeZO| 0.90 | 0.92 | 0.88 |
> | Ours|0.92 | 0.94 | 0.94 |
> | FO | 0.93 | 0.93 | 0.94 |

---

### Official Review · Reviewer_xRx2 · 2025-11-03

**Soundness:** 3
**Presentation:** 3
**Contribution:** 4
**Rating:** 6
**Confidence:** 3

**Summary:**

This paper proposes ZO Fine-tuner, a learning-based zeroth-order optimizer for fine-tuning large language models (LLMs).
Instead of using manually fixed perturbation strategies (e.g., MeZO), the authors design lightweight per-block neural networks (PertNNs) that learn adaptive perturbation variances.
The optimizer is trained once through a learning-to-learn (L2L) process on a single base model and can be reused across multiple downstream tasks or derived models.
Extensive experiments on 4 LLMs (1B–30B) and 7 datasets show consistent gains—82.1% of task–model pairs achieve lower loss than MeZO with <3.5% runtime overhead.

**Strengths:**

1. Introduces a learning-based perturbation policy for ZO optimizers—conceptually new compared to all existing handcrafted ZO methods.

2. Demonstrates strong generalization: trained once, the optimizer transfers to other datasets with consistent gains.

3. Theoretical justification (block-diagonal Hessian) is reasonable and empirically supported by ablations.

4. Runtime and memory overheads are negligible, maintaining MeZO-level efficiency.

Overall, I think this is a good paper.

**Weaknesses:**

1. The periodic reset policy, trajectory selection (optimizer/steps/lr), and seed sensitivity could materially affect the learned policy. While Algorithms 1–2 outline the process, the paper should report concrete reset schedules, ablate trajectory choices (e.g., Adam vs. SGD), and release fixed-seed scripts to ensure stable reproduction.

2. Hessian assumption not stress-tested across architectures.
The block-diagonal Hessian premise motivates block-wise variance. The paper would be stronger with empirical Hessian structure diagnostics or small-scale visualizations, especially for MoE or architectures with stronger cross-layer coupling.

**Questions:**

1. Can you provide head-to-head comparisons with the latest ZO optimizers targeting speed and variance reduction, using identical hardware, context length, and training budget?

2. How sensitive is the learned policy to the first-order trajectory source (AdamW vs. SGD, different lrs) used during L2L?

3. What is the reset schedule (trigger, frequency) in L2L, and how does it affect convergence and the loss-region coverage?

---

> ### Author Response · Authors · 2025-12-03
>
> We thank the reviewer for recognizing our contribution and greatly appreciate the feedback. We have addressed the reviewer’s concerns by providing more details on the L2L design and adding further ablation results. Below we provide a detailed response to each of the reviewer's concerns.
>
> > **W1 & Q2 & Q3: Questions about L2L details**
>
> **Trajectory choices.** We provide ablation results on the choice of first-order trajectory for LLaMA-3.2-1B and LLaMA-3.1-8B on SST-2 in Table 1 and Table 2. Across all intervals, Adam and SGD lead to very similar downstream loss curves and final test accuracy, and in some intervals SGD is slightly better. Since SGD has a significantly lower memory footprint than Adam, we use SGD for all reported L2L runs.
>
> **Periodic reset schedule.** We manually tune the reset schedule after inspecting the behavior of the first-order trajectory. For example, for LLaMA-3.2-1B, we observe that after 5 epochs of SGD, the loss on COPA drops to around 0.001. We therefore apply a reset every 5 epochs and train our ZO Fine-tuner for 15 epochs in total (with random shuffling of data). A similar tuning procedure is applied for all models.
>
> **L2L learning rates.** Since the L2L process involves multiple learning rates (for the first-order updates, the zeroth-order updates, and the ZO Fine-tuner itself), it is difficult to carry out an exhaustive ablation over all combinations. Our current learning-rate choices, detailed in Appendix C.3, are guided by earlier empirical testing, where we found that larger learning rates than those reported often lead to failure. We then fix these learning rates for all experiments and believe that our choice is already effective.
>
> **Importantly, although tuning the L2L process may appear intricate, it is done only once and then reused across datasets and model derivatives.** We also fully release our trained ZO Fine-tuner and fixed-seed scripts to reproduce the experiments in our paper.
>
> `Table 1: Ablation on first-order trajectory (SGD vs. Adam) on LLaMA 3.2-1B on SST2`
> |        | **Avg Loss [0,4k)** | **Avg Loss [4k,8k)** | **Avg Loss [8k,12k)** | **Avg Loss [12k,16k)** | **Avg Loss [16k,20k)** |
> |:--------:|:----:|:----:|-----------------------|------------------------|------------------------|
> | SGD    | 0.3732              | 0.2219               | 0.1855                | 0.1670                 | 0.1575                 |
> | Adam   | 0.3539              | 0.2213               | 0.1962                | 0.1808                 | 0.1655                 |
>
>
> `Table 2: Ablation on first-order trajectory (SGD vs. Adam) on LLaMA 3.1-8B on SST2`
>
> | | **Avg Loss [0,4k)** | **Avg Loss [4k,8k)** | **Avg Loss [8k,12k)** | **Avg Loss [12k,16k)** | **Avg Loss [16k,20k)** |
> |-|-|-|-|-|-|
> | SGD    | 0.4276              | 0.2758               | 0.2056                | 0.1702                 | 0.1527                 |
> | Adam   | 0.4159              | 0.2933               | 0.2303                | 0.1917                 | 0.1686                 |
>
> > **W2: Block-diagonal Hessian assumption**
>
> Computing Hessians for LLMs is challenging, even for relatively small models. Therefore, instead of reproducing full Hessian diagnostics ourselves, **we provide references to several works that observe block-structured Hessian behavior for standard Transformer architectures [1,2,3].** The models we study follow this standard Transformer architecture, and our block partition (embeddings, Q/K/V projections, output projections, MLPs) is aligned with these architectural components. Extending such Hessian analyses to more advanced architectures (e.g., MoE with stronger cross-layer coupling) is indeed interesting, and we **view this as an important direction for future work**.
>
> > **Q1: Baselines under the same training setting/budget**
>
> Our current experimental design already provides controlled, head-to-head comparisons among ZO optimizers that are specifically designed to reduce variance and/or improve speed within the MeZO family: HiZOO, LOZO, and MeZO-Adam/MeZO-AdamU. All methods are run on the same hardware, with the same context length and batch size. Moreover, as discussed in Appendix D.2, the additional Fine-tuner query introduces only negligible overhead, so the time complexity of ZO Fine-tuner is comparable to MeZO and about 1.5× faster than HiZOO. We therefore believe that the current experiments already demonstrate the effectiveness of our approach relative to these baselines under a similar computation budget.
>
> [1] Zhang, Yushun, et al. "Adam-mini: Use fewer learning rates to gain more." arXiv preprint arXiv:2406.16793 (2024).\
> [2] Zhang, Yushun, et al. "Why transformers need adam: A hessian perspective." Advances in neural information processing systems 37 (2024): 131786-131823.\
> [3] Ormaniec, Weronika, Felix Dangel, and Sidak Pal Singh. "What does it mean to be a transformer? insights from a theoretical hessian analysis." arXiv preprint arXiv:2410.10986 (2024).

---

### Author Response · Authors · 2025-12-03

We thank all the reviewers for their careful reading and constructive comments, and we thank the AC for their additional effort under this unusual situation. Below, we briefly summarize how we addressed the main concerns of each reviewer and where some issues stemmed from misunderstandings of our setting.

---

### **Reviewer xRx2**

**Main concerns:**

1. Details of the L2L hyperparameters.
2. Validity of the assumption that LLM Hessians are approximately block-diagonal.

**Our response:**

We addressed these points by:
1. Providing **more details of our L2L design**, including learning rates, trajectories, and reset schedules.
2. **Adding ablation studies on the first-order trajectory choice (SGD vs. Adam)**, showing similar downstream behavior and justifying the use of SGD for its lower memory footprint.
3. Clarifying that our **block-diagonal Hessian motivation is well-supported in the literature**, and that it serves as a heuristic motivation rather than a fragile assumption that the method critically depends on.

---

### **Reviewer 7BNG**

**Main concerns:**

The reviewer suggests comparisons against more memory-efficient ZO/FO baselines that use quantization, low-rank adaptation, or sparsity (e.g., QLoRA, SparseMeZO, SenseZOQ with quantization).

**Our response:**

We clarified that these directions are largely **orthogonal to the current focus of our work.** Our work's current focus is to improve the optimization quality of full-parameter ZO methods, not to compete with the most aggressively engineered memory-saving schemes. **In principle, our ZO Fine-tuner could also leverage sparsity, quantization, or a low-rank update, but we treat them as future works, rather than a fair comparison target for the present paper.**

---

### **Reviewer TJZo**

**Main Concerns:**

1. Dataset coverage.
2. The type of generalization demonstrated by the LLaMA 3.1-8B $\to$ LLaMA 3.1-8B-Instruct experiment.
3. The lack of accuracy curves (as opposed to only loss curves).

**Our response:**
1. We clarified that, in addition to short-sequence classification benchmarks, our experiments already include **benchmarks with substantially longer contexts (SQuAD and DROP)**. In the rebuttal, we **further added WiC and ReCoRD**, and our benchmark suite is consistent with those used in recent ZO optimization papers for LLMs.
2. We clarified that this experiment is designed to evaluate **within-family reuse**: a ZO Fine-tuner meta-trained on a base checkpoint (LLaMA 3.1-8B) can be directly reused on a derivative checkpoint (LLaMA 3.1-8B-Instruct), which is **practically important** given the current LLM supply-chain structure, where many deployed models are derivatives of a small number of base checkpoints.
3. We have now **added accuracy and F1 curves** (for SST-2 and SQuAD), which show that ZO Fine-tuner improves both convergence speed and final performance over MeZO.

---

### **Reviewer bS2A**

**Main Concerns:**

The reviewer questioned whether our main results are actually evaluated on held-out data.

**Our response:**

We clarified that this concern stems from a **big misunderstanding of our evaluation setup**. For all our experiments, **the ZO Fine-tuner is meta-trained only on COPA,** and all other benchmarks are unseen during L2L meta-training. Thus, the main results already evaluate generalization to unseen test datasets, rather than re-evaluating on the meta-training data.

---

### **Reviewer rsyj**

**Main Concerns:**

1. Potential additional biases in the method design beyond those already discussed.
2. Comparisons with parameter-efficient first-order baselines.

**Our response**

1. We emphasized that our design choices (e.g., blockwise structure, input features to PertNN) are **intentional compromises to make L2L tractable at LLM scale**, not arbitrary restrictions. Our **strong empirical results across models and datasets suggest that the method is practically effective**, even if the approximation is not perfect.
2. This concern overlaps with Reviewer 7BNG’s comments. Our opinion is the same: such baselines operate in a different regime (sparse/quantized/low-rank), while our work focuses on full-parameter ZO optimization. **We view combining ZO Fine-tuner with these techniques as promising future work, rather than being within the scope of this paper.**

---

### Meta-Review · Area_Chair_2533 · 2026-01-03

**Summary:**

This paper proposes ZO Fine-tuner, a learning-to-learn (L2L) zeroth-order optimizer for fine-tuning large language models. Rather than relying on hand-crafted perturbation schemes (e.g., MeZO), the method learns block-wise adaptive perturbation variances via lightweight auxiliary networks. The optimizer is trained once per base model and reused across downstream tasks and derivative checkpoints.

Reviewers generally agreed that the paper is technically sound, well implemented, and empirically validated across multiple LLMs and benchmarks, demonstrating consistent improvements over existing zeroth-order baselines with negligible overhead. However, several concerns influenced the final assessment: (i) the scope and generality of the “train once, reuse widely” claim, (ii) the absence of comparisons to stronger memory-efficient FO/ZO baselines, and (iii) limited theoretical justification, particularly regarding potential bias from anisotropic perturbations and the block-diagonal Hessian assumption. Reviewer scores spanned from marginal accept to reject, reflecting a divide between appreciation of the empirical results and reservations about conceptual depth.

Overall, the average score remains below the acceptance threshold, and no clear consensus in favor of acceptance emerged. While the work is solid and carefully executed within its stated scope, reviewers generally viewed the contribution as incremental rather than foundational, with unresolved theoretical questions and limited evaluation against stronger practical baselines.

**Reviewer Concerns:**

Concerns largely addressed by the rebuttal:

•	Generalization across tasks and held-out data:
The authors clarified that meta-training is performed only on COPA, and all other datasets are strictly unseen at training time. Additional results on ReCoRD and WiC further support cross-task generalization.
•	L2L design details and reproducibility:
Reviewers’ concerns about reset schedules, trajectory choices, and optimizer selection were addressed with new ablations (SGD vs. Adam), detailed explanations, and a commitment to releasing fixed-seed scripts.
•	Lack of accuracy curves:
The authors added accuracy and F1 curves (e.g., SST-2, SQuAD), showing consistent improvements over MeZO in both convergence speed and final performance.


Concerns that remain partially or fully outstanding:

•	Overstated generality claims:
While within-family reuse (e.g., LLaMA-8B → LLaMA-8B-Instruct) is convincingly demonstrated, several reviewers remain unconvinced that this supports broader cross-model generalization.
•	Theoretical weaknesses:
Concerns about directional bias from anisotropic perturbations, greedy one-step meta-objectives, and teacher-forcing effects during meta-training were acknowledged but not fully resolved.
•	Comparisons to stronger memory-efficient baselines:
Although the authors correctly argue these are orthogonal, multiple reviewers felt that the lack of LoRA/QLoRA or sparse-ZO comparisons limits the paper’s practical positioning.

**Reviewer Scores:**

•	Reviewer xRx2: Likely remains at 6 (Weak Accept) given strong empirical validation and addressed concerns.
•	Reviewer 7BNG: Likely remains at 4 (Borderline Reject); scope and baseline concerns persist.
•	Reviewer TZJo: Likely improves from 2 → 3 (Borderline Reject) after added datasets and accuracy curves.
•	Reviewer bS2A: Likely improves from 2 → 3, as the generalization misunderstanding was clearly resolved.
•	Reviewer rsyj: Likely remains at 4, with theoretical concerns still largely unaddressed.

---

### Decision · Program_Chairs · 2026-01-26

Reject